# Longitudinal genomic analysis of *Neisseria gonorrhoeae* transmission dynamics in Australia

Mona L. Taouk [1,2], George Taiaroa[1,2], Sebastian Duchene [3,4], Soo Jen Low [1,2], Charlie K. Higgs[3], Darren Y. J. Lee[3], Shivani Pasricha[1,5], Nasra Higgins[6], Danielle J. Ingle [3], Benjamin P. Howden [3,7], Marcus Y. Chen[8,9], Christopher K. Fairley[8,9], Eric P. F. Chow [8,9,10] & Deborah A. Williamson [1,11,12] ✉

*N. gonorrhoeae*, which causes the sexually transmissible infection gonorrhoea, remains a significant public health threat globally, with challenges posed by increasing transmission and antimicrobial resistance (AMR). The COVID-19 pandemic introduced exceptional circumstances into communicable disease control, impacting the transmission of gonorrhoea and other infectious diseases. Through phylogenomic and phylodynamic analysis of 5881 *N. gonorrhoeae* genomes from Australia, we investigated *N. gonorrhoeae* transmission over five years, including a time period during the COVID-19 pandemic. Using a novel cgMLST-based genetic threshold, we demonstrate persistence of large *N. gonorrhoeae* genomic clusters over several years, with some persistent clusters associated with heterosexual transmission. We observed a decline in both *N. gonorrhoeae* transmission and genomic diversity during the COVID-19 pandemic, suggestive of an evolutionary bottleneck. The longitudinal, occult transmission of *N. gonorrhoeae* over many years further highlights the urgent need for improved diagnostic, treatment, and prevention strategies for gonorrhoea.

*Neisseria gonorrhoeae*, the causative agent of gonorrhoea, is the second most common sexually transmitted infection (STI) worldwide, with approximately 82 million new infections in 2020[1]. In Australia, the incidence of gonorrhoea notifications has increased markedly over the last decade, with an increase of 105.7% between 2012 (61.5 notifications/100,000) and 2022 (126.5 notifications/100,000)[2]. Untreated,

gonorrhoea can result in severe sequelae such as pelvic inflammatory disease, infertility, and ectopic pregnancy, and may promote the acquisition and transmission of HIV[3–5]. As gonorrhoea is frequently asymptomatic, particularly for women and at extragenital infections, diagnosis is often achieved through asymptomatic screening, with a considerable reservoir of infection in the community[6,7]. Further,

[1]Department of Infectious Diseases, The University of Melbourne at the Peter Doherty Institute for Infection and Immunity, Melbourne, VIC, Australia. [2]Victorian Infectious Diseases Reference Laboratory, The Royal Melbourne Hospital at The Peter Doherty Institute for Infection and Immunity, Melbourne, VIC, Australia. [3]Department of Microbiology and Immunology, The University of Melbourne at the Peter Doherty Institute for Infection and Immunity, Melbourne, VIC, Australia. [4]Department of Computational Biology, Institut Pasteur, Paris, France. [5]Infectious Diseases and Immune Defence Division, Walter and Eliza Hall Institute of Medical Research, Parkville, Melbourne, Australia. [6]Victorian Department of Health, Melbourne, VIC, Australia. [7]Microbiological Diagnostic Unit Public Health Laboratory, The University of Melbourne at the Peter Doherty Institute for Infection and Immunity, Melbourne, VIC, Australia. [8]Melbourne Sexual Health Centre, Alfred Health, Melbourne, VIC, Australia. [9]School of Translational Medicine, Faculty of Medicine, Nursing and Health Sciences, Monash University, Melbourne, VIC, Australia. [10]Centre for Epidemiology and Biostatistics, Melbourne School of Population and Global Health, The University of Melbourne, Melbourne, VIC, Australia. [11]Department of Medicine, University of St Andrews, St Andrews, Fife KY16 9TF, Scotland. [12]MRC-University of Glasgow Centre for Virus Research, Glasgow G61 1QH, Scotland. ✉e-mail: deborah.williamson@unimelb.edu.au

antimicrobial resistance (AMR) in *N. gonorrhoeae* is a major public health concern, with the evolution of strains carrying resistance to all therapeutic antimicrobial classes[8–12]. The emergence and spread of AMR strains together with increasingly limited antimicrobial options complicates empiric therapy, with a need for ongoing AMR surveillance to inform standard treatment guidelines.

The COVID-19 pandemic resulted in rapid, extraordinary, and sweeping changes to society. During the pandemic, public health measures requiring physical distancing, social isolation, and stay at home measures changed societal behaviours and the delivery of healthcare services at local, national, and global levels. Although the aim of social distancing was to prevent the spread of SARS-CoV-2, the causative agent of COVID-19, these measures impacted the transmission and spread of other close-contact infectious diseases, including STIs[13–16]. For example, 'stay at home' directions and closure of social venues reduced opportunities for casual sexual encounters ('sexual distancing'), reducing STI transmission[17,18]. Further, interruption of sexual and public health services may have delayed STI diagnoses, negatively impacting control of transmission as reported in countries like the United States[19,20]. In Australia, an initial wave of COVID-19 occurred from March 2020, resulting in national public health restrictions, including closing of the international border to all non-citizens on 20th March, leading to state-wide social restrictions[21]. For the state of Victoria, an initial lockdown from 31st March 2020 to 12th May 2020 was followed by a 'second wave' of COVID cases, leading to a lockdown from 9th July 2020 to 27th October 2020 and four subsequent lockdowns across 2021[22,23]. National notification disease surveillance (NNDSS) data from Australia showed a reduction in STIs, including gonorrhoea, in early 2020 coinciding with national COVID-19 public health restrictions[24]. Public health measures in Victoria were some of the most stringent globally and the impact of social distancing on gonorrhoea transmission has not been thoroughly studied. Assessment of incidence alone as a marker of gonorrhoea transmission does not provide information on the underlying sexual networks contributing to persistent chains of transmission[25]. Understanding these factors is key to determining the contemporary drivers of gonorrhoea transmission, contributing to more effective prevention and control strategies.

In a previous study we integrated *N. gonorrhoeae* whole genome sequencing (WGS) data with demographic, epidemiological and behavioural data to detail *N. gonorrhoeae* transmission in Melbourne, Australia throughout 2017[25]. This previous study revealed transmission within and between population groups with differing epidemiological risk factors including gender, sexual behaviour, overseas travel, HIV status and pre-exposure prophylaxis (PrEP) use. Although major circulating lineages were identified, each with varying patterns of AMR, there remains limited data on the specific individual and pathogen factors that may drive persistence of transmission networks over several years. Accordingly, we carried out a longitudinal genomic study, defining *N. gonorrhoeae* putative transmission networks over a four-and-a-half-year period. Further, we assess the effects of COVID-19 social distancing measures on *N. gonorrhoeae* genomic diversity and chains of transmission.

## Results
### Epidemiological characteristics of the dataset
Between 1st January 2017, and 30th June 2021, there were 31,187 gonorrhoea notifications in Victoria, Australia (Supplementary Fig. 1). Of these notifications, 24,212 (77.6%) were from males and 6739 (21.6%) were from females, with 236 of unknown gender. In total, 6329 *N. gonorrhoeae* isolates underwent WGS and 6215 (98.2%) passed quality control cheques. 5,608 genomes represented unique infections collected from one individual each. The dataset also included 558 genomes collected from different body sites from 273 individuals (within-individual samples). A further 49 genomes were colony picks which were all collected from 13 individuals in total (within-site samples) (Supplementary Fig. 2, Supplementary Data 1). In total, 5881 genomes (one samples collected per individual, per day; this was randomly selected for within-individual samples) were included for clustering and epidemiological analyses making up 18.9% of all gonorrhoea notifications in Victoria were for this period (Supplementary Data 1).

Most isolates were from a urogenital site (3136/5881; 53.3%), followed by rectal sites (1702/5881; 29.0%) and then pharyngeal sites (975/5881; 16.6%) (Table 1). Consistent with notification data, most individuals with *N. gonorrhoeae* isolates were male (5051/5881; 86.0%). Of those males with reported sexual behaviour data, 2206/2410 (91.5%) identified as gay, bisexual, or other men who have sex with men (GBMSM) which included men who have sex with men only (MSMO) and men who have sex with men and women (MSMW) (Table 1). The median age of individuals was 30 years (interquartile range: 26–38); with 42.0% (2472/5881) individuals in the 20–29 range group (Table 1).

### Genotypic and phenotypic characteristics of N. gonorrhoeae isolates
All isolates underwent phenotypic antimicrobial susceptibility testing (Supplementary Table 1). Phenotypic resistance was observed to penicillin (1661/5881; 28.2%), tetracycline (3624/5881; 61.6%), ciprofloxacin (2136/5881; 36.3%) and azithromycin (390/5881; 6.6%), and while there were no ceftriaxone resistant isolates, several isolates (95/5881; 1.6%) displayed decreased susceptibility to ceftriaxone. Of all isolates, 68.9% (4052/5881) were resistant to at least one antimicrobial. Further, of the 4015 isolates resistant to one of penicillin, tetracycline, or ciprofloxacin, 1179 (29.4%) were resistant to all three (Supplementary Fig. 3).

There were 135 unique multi-locus sequence types (MLSTs), with the most common being ST8156 (1279/5881; 21.7%), ST7363 (738/5881; 12.5%), and ST11864 (668/5881; 11.4%). 54/5881 (0.9%) of genomes in the dataset were typed as a unique MLST each (Supplementary Fig. 4). The most common NG-STAR profile was ST442 (1186/5881; 20.2%) (*penA*: Type II NonMosaic; A517G, *mtrR*: −35A Del, *ponA*: L421P). The

## Table 1 | Characteristics of individuals with *Neisseria gonorrhoeae* isolates included in this study

| Characteristic | Number/Total (%) |
|---|---|
| **Sex** | |
| Male | 5060/5881 (86.0) |
| GBMSM | 2206/5060 (43.6) |
| MSW | 204/5060 (4.0) |
| Not reported | 2650/5060 (52.4) |
| Female | 776/5881 (13.2) |
| Other | 45/5881 (0.8) |
| Not reported | 19/5881 (0.3) |
| **Isolation site** | |
| Pharyngeal | 975/5881 (16.6) |
| Rectal | 1702/5881 (29.0) |
| Urogenital | 3136/5881 (53.3) |
| Other | 40/5881 (0.7) |
| Not reported | 28/5881 (0.5) |
| **Age group** | |
| 0–19 years | 182/5881 (3.1) |
| 20–29 years | 2472/5881 (42.0) |
| 30–39 years | 1951/5881 (33.2) |
| 40–49 years | 789/5881 (13.4) |
| Over 50 years | 487/5881 (8.3) |

*BGMSM* Gay, bisexual, and other men who have sex with men.
*MSW* Men who have sex with women.

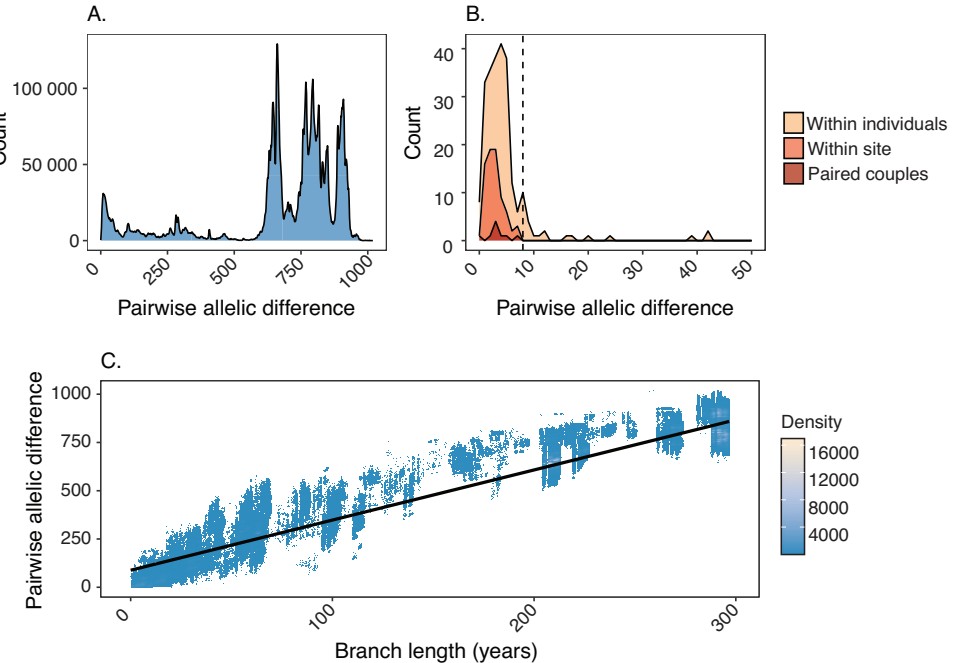

**Fig. 1 | Distribution of the number of cgMLST allelic differences. A** Distribution of the number of cgMLST allelic differences between pairs of isolates among the 5881 genomes collected from different individuals (between individual isolates). **B** Distribution of the number of cgMLST allelic differences between pairs of isolates among the 273 within-individual isolates, 49 colony pick within-site samples and the eight pairs of deidentified sexual partners, with the x-axis limited to 50 allelic differences. The dashed vertical line represents the threshold value for inclusion within a transmission cluster (7 allelic differences). **C** A nomogram showing allelic difference plotted against the branch distance for every pair of isolates included in the study. Branch lengths were extracted from a maximum likelihood phylogeny of 5881 isolates and dated using least-square dating to estimate the rate of evolution of the phylogeny. A linear regression line is shown in black.

second most prevalent NG-STAR was ST439 (598/5881; 10.2%) (*penA*: Type II NonMosaic; A517G, *mtrR*: A39T). The third most prevalent NG-STAR was ST231 (327/5881; 5.6%) (*penA*: Type II NonMosaic; A517G, *mtrR*: A39T). There were 270 NG-STAR profiles defined in the dataset overall (Supplementary Fig. 4). 3/5881 (0.05%) of isolates could not be assigned an NG-STAR profile. Additionally, 543/5881 (9.2%) of isolates were identified as one of 156 new NG-STAR profiles (Supplementary Data 1).

**Defining N. gonorrhoeae transmission groups**
To define putative transmission groups, a cgMLST-based approach was applied. The available cgMLST scheme of 1648 genes for *N. gonorrhoeae* was adapted using a threshold of 95%, such that only genes present in at least 95% of the dataset were included in the allele calling process to create a core-cgMLST scheme for this dataset (Supplementary Fig. 5)[26]. This resulted in 1495 genes in the core-cgMLST scheme, with a maximum, median and mean pairwise allelic difference of 1019, 765 and 683.6 across the dataset, respectively (Fig. 1A). Following adaptation of the cgMLST scheme, a mean of 99.9% (range: 98.2% to 100%) of loci were designated an allele for each isolate. To determine an appropriate genetic distance threshold to define genomic clusters, pairs of isolates were grouped as: (i) between individual (any two isolates collected from different individuals); (ii) within individual (any two isolates collected from the same individual at different body sites); (iii) within site (any two isolates collected from the same individual from the same body site) or (iv) paired couples (pairs of sexual partners identified from case data) (Supplementary Figs. 6–7). The median pairwise allelic difference between individuals, within individuals, within sites and in paired couples was 765 (range: 0–1019), 4 (range: 0–966), 3 (range: 0–7) and 3 (range: 0–7), respectively (Fig. 1B).

Using a time scaled phylogeny, the molecular evolution of *N. gonorrhoeae* in this dataset was estimated to be $4.995 \times 10^{-2}$ allelic differences per week (95% credibility interval: $4.994 \times 10^{-2}$ – $5.000 \times 10^{-2}$), or approximately three allelic differences per year (Fig. 1C; Supplementary Figs. 8–9). Genomic clusters were defined using the corrected pairwise differences for single linkage clustering and applying a threshold of 7 allelic differences, taken from the maximum pairwise allelic difference between any two within-site or paired couple isolates (Supplementary Figs. 10–11; Supplementary Table 2).

Amongst the 5881 *N. gonorrhoeae* isolates representing a unique infection each, 233 genomic clusters ($n \geq 3$, 4720/5881, 80.3% genomes), 151 pairs ($n = 2$; 302/5881, 5.1% genomes) and 878 singletons ($n = 1878/5881$, 14.9%) were identified (Fig. 2). Pairs and clusters made up 85.4% (5022/5881) of the dataset. The median cluster size was 6 individuals (range: 3–709 individuals) and the median persistence of groups was 25.7 weeks (range: 0–231.9 weeks). Eleven putative genomic clusters spanned the entire sampling timeframe (1st January 2017 to 30th June 2021) ranging in size from 32 isolates to 709 isolates.

**Demographic associations with persistent gonorrhoea clusters**
To identify potential demographic associations with large genomic clusters of *N. gonorrhoeae* (comprising groups with ≥ 30 individuals), epidemiological and genomic data were integrated (Fig. 3). In total, 31 large genomic clusters were identified with a median cluster size of 59 individuals, ranging from 31 (Cluster 16) to 709 individuals (Cluster 180) with 3226 individuals included collectively. All large genomic clusters each comprised a single MLST (excluding genomes that could not be assigned an MLST), with the most common being ST8156 ($n = 5$ clusters) and ST7363 ($n = 4$ clusters) (Fig. 3).

12/31 (38.7%) of large genomic clusters (≥ 30) were defined as persistent (time between the earliest sample collection and latest

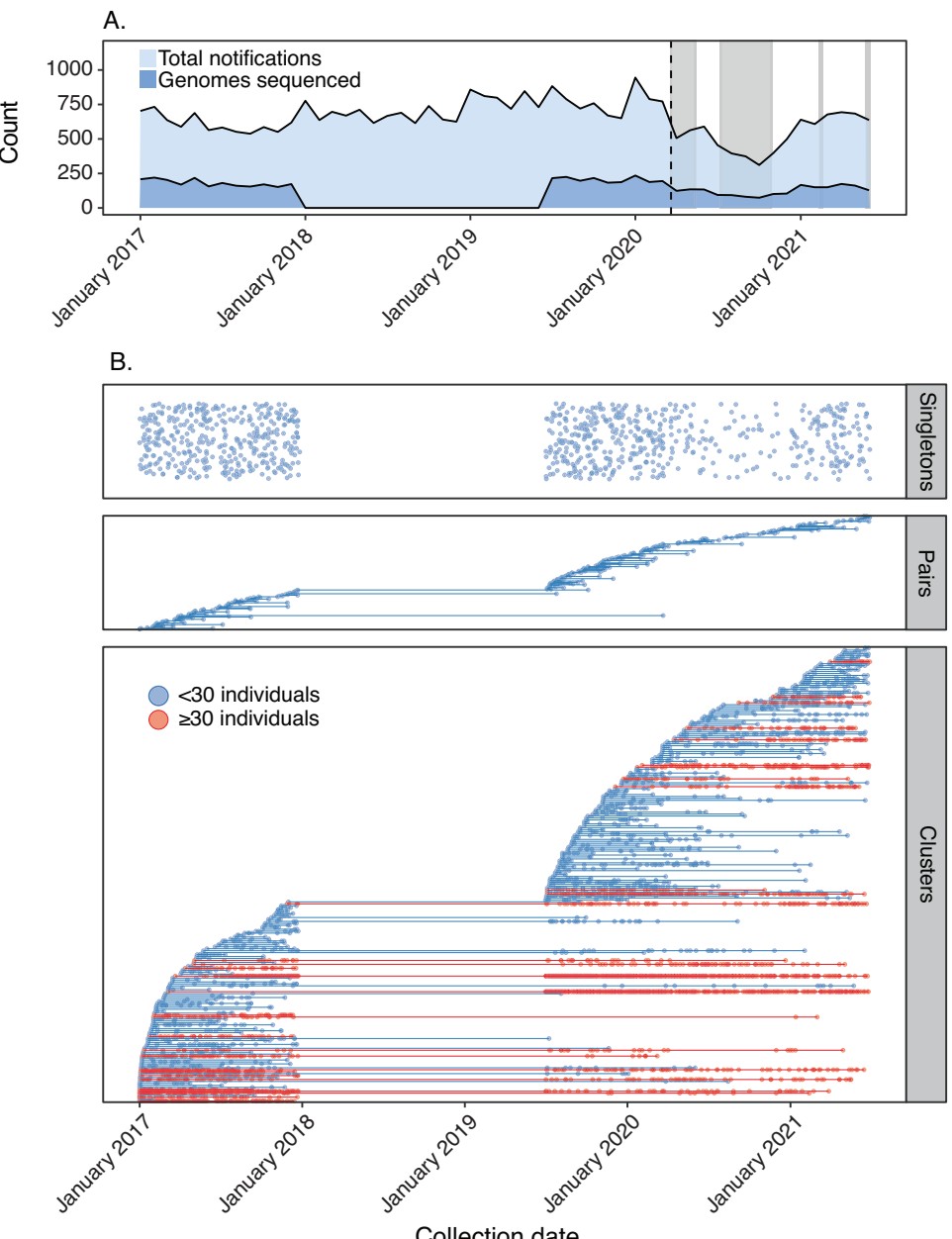

**Fig. 2 | Temporal distribution of clusters, pairs and singletons. A** Notifications of gonorrhoea infections in Victoria between January 2017 and June 2021 (inclusive). The proportion of genomes sequenced out of total notifications is shown. The dashed vertical line represents the date of the Australian international border closure (20th March 2020). The grey regions represent social restrictions within the state of Victoria. **B** Clusters of Neisseria gonorrhoeae genomes in Victoria, Australia, 2017. Genomes are classified according to whether they are single isolates, pairs (2 isolates) or in a cluster ( ≥ 3 isolates). Each pair or cluster contains all individuals with isolates related by ≤ 7 pairwise allelic differences using single-linkage hierarchical clustering. Each dot represents an isolate from an individual, and each cluster is plotted along a horizontal line, representing date of sample collection.

sample collection > 2 years) (Fig. 3). Additionally, for 10/202 (5.0%) small clusters (< 30 individuals) the date between earliest sample collection and latest sample collection was > 2 years (range: 126.9–219.4 weeks) (Supplementary Fig. 12). Multivariable logistic regression analysis showed that larger clusters were significantly more likely to persist (adjusted odds ratio (aOR) 1.02; 95% CI 1.00–1.03; $P = 0.03$), and females were significantly more likely to be part of a persistent cluster compared to males (aOR 2.91; 95% CI 1.21–7.00; $P = 0.017$) (Table 2). However, there were no significant associations between age of individuals in a genomic cluster and persistence (Table 2). Additionally, we found no significant associations between N. gonorrhoeae that were phenotypically resistant to penicillin,

tetracycline, or ciprofloxacin with persistence (Table 2). A significant association was observed for N. gonorrhoeae isolates characterised as phenotypically azithromycin susceptible, however, with these more likely found in persisting genomic clusters (aOR 126; 95% CI 12.9–1238; $P < 0.0001$) (Table 2). Although this observation was driven by Clusters 1 and 3 which were non-persisting and resistant (Fig. 3).

### Neisseria gonorrhoeae genomic diversity and transmission during the COVID-19 pandemic

Gonorrhoea notifications in Australia declined from early to mid 2020, contemporaneous with nationwide COVID-19 public health restrictions (Supplementary Fig. 1). To assess changes in genomic diversity

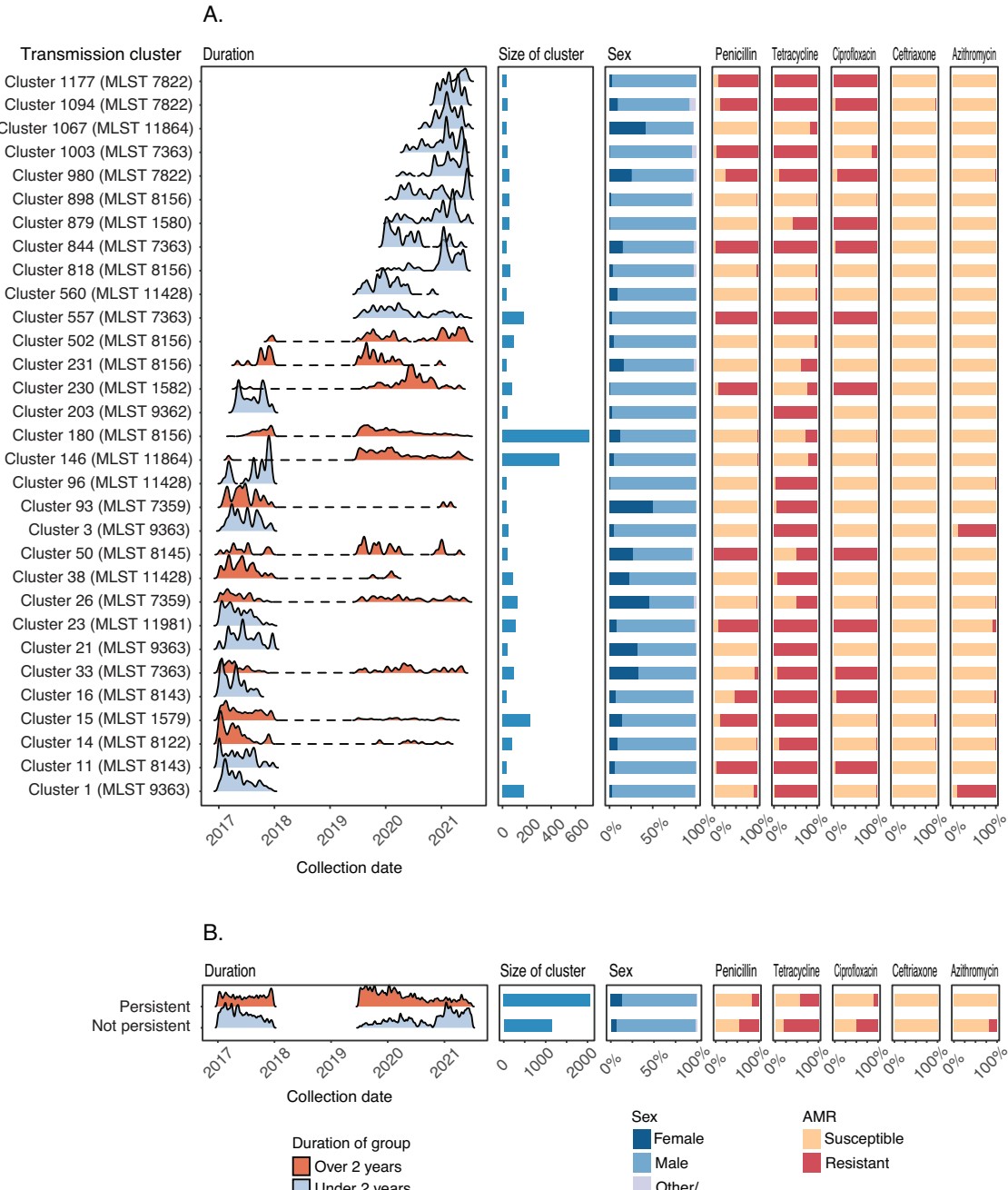

**Fig. 3 | Associations of large Neisseria gonorrhoeae clusters with epidemiological factors and antimicrobial susceptibility profiles.** Large clusters are defined as those containing 30 or more individuals. **A** Temporal distribution collection date for large clusters. Clusters are coloured as persistent (duration over two years) or non-persistent (duration under two years). Plots shown include, the number of isolates grouped in each cluster, the percentage of male, female, or other individuals in each cluster and the phenotypic antimicrobial susceptibility profiles of isolates in each cluster for each of penicillin, tetracycline, ciprofloxacin, ceftriaxone, and azithromycin. For all plots except ceftriaxone, "susceptible" includes isolates characterised as "intermediate", "less susceptible" or "decreased susceptibility". **B** The temporal distribution of isolates separated by whether they belong to a persistent or non-persistent cluster. Plots shown include, the number of isolates grouped in each cluster, the percentage of male, female, or other individuals in each cluster and the phenotypic antimicrobial susceptibility profiles of isolates in each cluster for each of penicillin, tetracycline, ciprofloxacin, ceftriaxone, and azithromycin. For all plots except ceftriaxone, "susceptible" includes isolates characterised as "intermediate", "less susceptible" or "decreased susceptibility".

over time, the median count of unique alleles per cgMLST core gene prior to the COVID-19 pandemic (1st July 2019 to 31st March 2020 inclusive; $n = 2489/3714$) and after (1st April 2020 to 30th June 2021 inclusive; $n = 1225/3714$) was quantified. A decline in the median count of unique alleles per gene from 11 in pre-Covid restrictions data to 8 during Covid restrictions ($p$-value $< 0.0001$; Mann–Whitney Rank sum

test) was observed indicating a decline in genomic diversity (Fig. 4A). Despite an increase in gonorrhoea notifications during 2021, reduced genomic diversity persisted throughout the study period (Fig. 4A).

To quantify the effect of COVID-19 restrictions on transmission, we applied a hierarchical Bayesian phylodynamic framework to calculate the effective reproductive number ($R_e$) for clusters before and

**Table 2 | Associations between persistence of clusters and individual demographic factors, size of cluster, and phenotypic antimicrobial resistance**

| Characteristic | Adjusted Odds ratio | 95% Confidence Interval | P-value | n/N | % |
|---|---|---|---|---|---|
| **Sex** | | | | | |
| Male | Reference | - | - | 1760/2805 | 62.7 |
| Female | 2.910 | 1.21–7.00 | 0.017 | 284/367 | 77.4 |
| Other | 0.749 | 0.189–2.97 | 0.681 | 17/30 | 56.7 |
| Unknown | 0.111 | 0.00289–4.290 | 0.239 | 1/6 | 16.7 |
| **Age** | | | | | |
| 16–19 years | Reference | - | - | 69/100 | 69.0 |
| 20–29 years | 0.682 | 0.316–1.470 | 0.328 | 816/1270 | 64.3 |
| 30–39 years | 0.616 | 0.318–1.190 | 0.151 | 695/1087 | 63.9 |
| 40–49 years | 0.483 | 0.197–1.190 | 0.113 | 319/493 | 64.7 |
| 50 years and older | 0.458 | 0.171–1.230 | 0.121 | 163/258 | 63.2 |
| **Size** | | | | | |
| Size | 1.020 | 1.000–1.030 | 0.030* | - | - |
| **Phenotype** | | | | | |
| **Penicillin** | | | | | |
| Susceptible/less susceptible | 2.71 | 0.249–29.500 | 0.413 | 1743/2371 | 73.5 |
| Resistant | Reference | - | - | 319/837 | 38.1 |
| **Tetracycline** | | | | | |
| Susceptible/intermediate | 1.800 | 0.292–11.100 | 0.525 | 1190/1411 | 84.3 |
| Resistant | Reference | - | - | 872/1797 | 48.5 |
| **Ciprofloxacin** | | | | | |
| Susceptible/less susceptible | 2.620 | 0.320–21.200 | 0.368 | 1838/2409 | 76.3 |
| Resistant | Reference | - | - | 224/799 | 28.0 |
| **Ceftriaxone** | | | | | |
| Susceptible | 0.130 | 0.016–1.060 | 0.057 | 2050/3195 | 64.2 |
| Decreased susceptibility | Reference | - | - | 12/13 | 92.3 |
| **Azithromycin** | | | | | |
| Susceptible | 126 | 12.900–1238 | 0.0000326 * | 2052/2991 | 68.6 |
| Resistant | Reference | - | - | 10/217 | 4.6 |

Coefficients with p-values < 0.05 are noted with * as calculated using a two sided likelihood ratio test. n/N refers to the number of persistent isolates in each subcategory out of the total isolates in the subcategory. % is n/N expressed as a percentage. For AMR phenotype, isolates were grouped binarily as either resistant or not resistant except for ceftriaxone where there were no resistant isolates.

after the implementation of social distancing (see methods; Supplementary Fig. 13). For clusters with at least 5 genomes, we observed a decrease in average effective reproductive number ($R_e$) from late March 2020 (95% credible interval between the 20th and 24th of March) coinciding with the implementation of public health measures. The mean $R_e$ prior to the implementation of public health measures was 1.46 (95% credible interval: 1.38–1.55), while following the implementation of public health measures it was 1.26 (95% credible interval: 1.13–1.39) (Fig. 4B, C). The posterior probability of a decrease in the mean $R_e$ after public health measures was 0.99 (corresponding Bayes factor support of 95 or 'decisive' evidence). Our inference of heterogeneity in $R_e$ among clusters was substantially higher after public health measures, implying that a small number of clusters were likely driving transmission (Fig. 4D, Supplementary Fig. 14).

## Discussion

Gonorrhoea remains a significant public health concern, with challenges posed by increasing incidence and the continual emergence of AMR[8–12]. Here, we provide a large-scale genomic investigation into the longitudinal transmission of *N. gonorrhoeae*. We provide insights into the transmission dynamics through a period encompassing the implementation of public health measures (social distancing, lockdowns, closure of international and state borders) aimed at controlling COVID-19 in Australia during 2020 and 2021[22,23]. Like other studies, we observed a decline in gonorrhoea notifications concurrent with

COVID-19 restrictions. We also demonstrate reduced genomic diversity of circulating *N. gonorrhoeae*, suggesting a temporary disruption in transmission networks[27]. Reduced genomic diversity among *N. gonorrhoeae* isolates persisted even as case numbers increased once restrictions were lifted. It is not known if this reduction in genomic diversity will continue to impact gonorrhoea transmission dynamics over subsequent years.

As shown previously, the integration of genomic data with epidemiological and behavioural information remains a powerful tool for identifying transmission networks, including for sexually transmitted diseases[25,28]. Recently, there has been considerable growth and development in related fields, such as for phylogenetic inferences, although many of the methods remain reliant on strict core genome alignments typically after masking for recombination[29]. These methods have known limitations for highly recombinogenic or genetically diverse species, such as *N. gonorrhoeae*, with the use of a strict core genome alignment potentially resulting in isolates being classified as more closely related than they would be using other bioinformatic approaches[30,31]. In this study, we applied methods of clustering and inferring transmission which have been established recently and address these limitations, including the use of an adapted cgMLST scheme in *N. gonorrhoeae*[26,30]. Here, we adapted a previously defined cgMLST scheme, initially created to assess the large-scale population structure of the species, using a high-resolution threshold to describe genomic clusters[26]. The rationale for selecting a threshold for defining

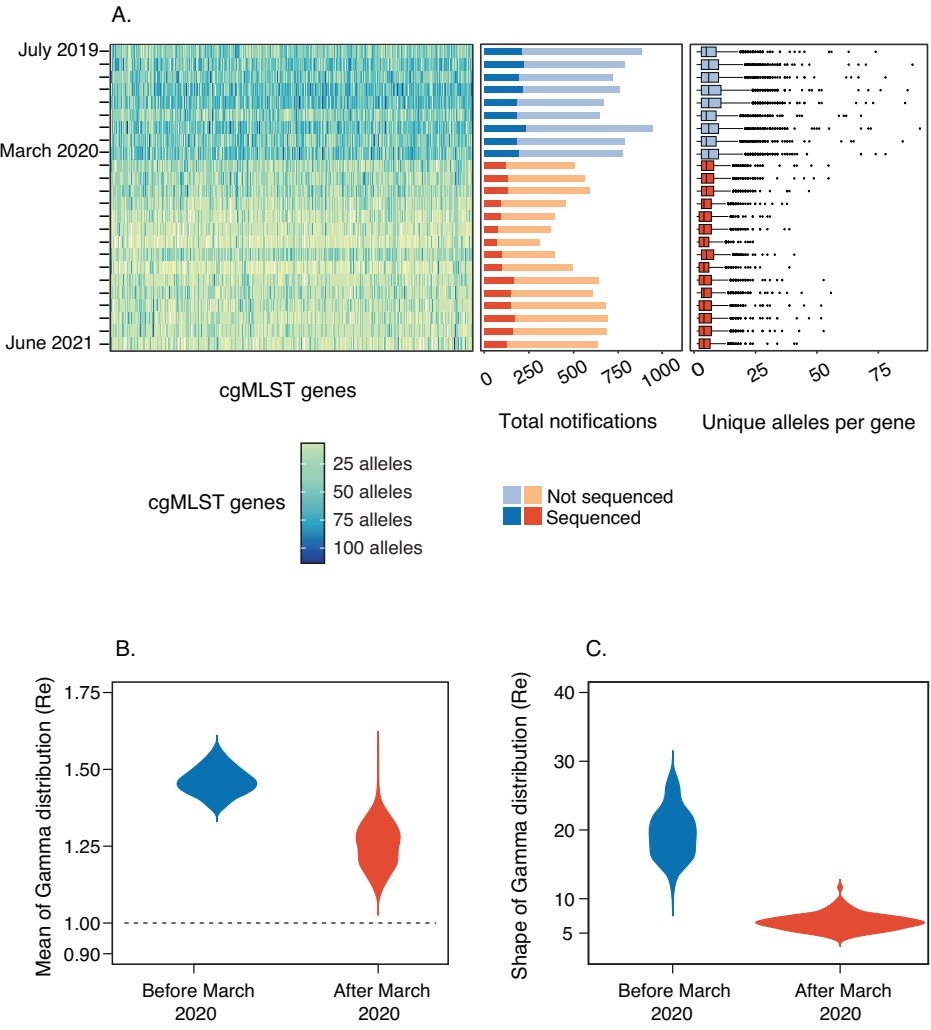

**Fig. 4 | Diversity and effective reproductive number estimates. A** Heatmap that represents the number of unique alleles typed at each gene of the cgMLST scheme for each one-month interval of the time frame. The lightest colour represents the minimum number of alleles, and the darkest colour represents the maximum number of alleles for each gene. The bar chart represents the total notifications of gonorrhoea in the state and the proportion that was sequenced for each month. The boxplot represents the number of unique alleles for each of the 1495 genes in the scheme for each month. Box plots indicate median and interquartile range (IQR), with the whiskers representing the highest and lowest values within 1.5 × IQR of the upper and lower quartiles, and the dots representing outlier values. The dataset includes months prior to non-pharmaceutical interventions (1st July

2019–31st March 2020) and following non-pharmaceutical interventions (1st April 2020–30th June 2021), noting that isolates collected in 2017 were not included in this analysis. Violin plots for posterior densities for the parameters of the Gamma distribution of the reproductive number ($R_e$) before and after the time slice (coinciding with introduction of lockdowns in Victoria). In (**B**) we show the mean distribution of mean $R_e$, for each of the 97 transmission clusters with the dashed line corresponding to a value of 1.0. In (**C**) we show the shape of the Gamma distribution for each of the 97 transmission clusters that represents the contribution of clusters to $R_e$, with smaller values indicating that a small number of clusters contribute to spread.

genomic clusters based on within-site isolates and epidemiologically linked pairs, and correction for evolutionary distance further underpins the robustness of our study's framework and enhances the reliability of genetic relationships drawn from the data.

Our finding of large and temporally persistent genomic clusters, some spanning the entire study period, demonstrates the existence of stable chains of transmission persisting in the population for years, and provides insights into the demographic and microbiological factors that may contribute to the stability of *N. gonorrhoeae* transmission networks. Our observation that clusters with a higher ratio of women to men were significantly more likely to persist for longer periods of time, suggest that chains of transmission occurring amongst heterosexual populations may persist for longer periods than chains of transmission in GBMSM communities in our setting. Additionally, gonorrhoea is more likely to be asymptomatic in women than men, potentially resulting in a delayed diagnoses that may promote

transmission and the persistence of genomic clusters[6]. An increase in gonorrhoea diagnoses among women in Victoria has been observed over the past few years, predating the COVID-19 pandemic, and coinciding with a reported rise in syphilis cases among women in the state[32]. Previous research has shown that women with syphilis are disproportionately represented among socioeconomically disadvantaged women[33]. It is plausible that the parallel rise in gonorrhoea notifications among women, as well as the finding of persistent genomic clusters containing women, may highlight limited access to testing and treatment in disadvantaged populations. These findings highlight the need for targeted interventions to address the specific needs and behaviours of at-risk populations.

In addition, the presence of persistent *N. gonorrhoeae* transmission raises significant concerns regarding the demonstrated ability of *N. gonorrhoeae* to develop AMR, with stable genomic clusters potentially acting as reservoirs for AMR strains[34]. The

recommended management of genital and anorectal gonorrhoea infections in Australia is a dual therapy of Ceftriaxone 500 mg and Azithromycin 1 g (Azithromycin 2 g for pharyngeal infection)[35]. The absence of significant associations between phenotypic resistance to penicillin, tetracycline, or ciprofloxacin and persistence suggests that AMR may not be a primary driver of persistence in this setting particularly as these antimicrobials are not currently recommended for therapeutic treatment in Australia. Susceptibility to azithromycin, a current recommended therapeutic, was associated with persistence, further highlighting the complex interplay between antimicrobial susceptibility and transmission. This finding aligns with existing research demonstrating a fitness cost associated with antibiotic resistance. For instance, cefixime-resistant strains of *N. gonorrhoeae* have been shown to cause approximately half the number of secondary infections compared to susceptible strains[36]. Similarly, azithromycin-resistant *Staphylococcus aureus* strains exhibit reduced fitness compared to azithromycin-sensitive strains[37]. These findings collectively suggest that resistance may incur a fitness disadvantage, which could reduce the transmission potential of resistant strains and that alternate factors such as host-pathogen interactions or behavioural dynamics may play more central roles in persistent chains of transmission.

The application of a hierarchical Bayesian phylodynamic framework to quantify changes in bacterial transmission over time represents a novel approach to understanding the dynamics of gonorrhoea emergence and spread[38]. Our investigation revealed a decline in transmission of *N. gonorrhoeae* during 2020 and 2021, a period characterised by reduced social interactions and stringent national and international border closures in Australia. Consistent with other findings that indicate a decline in the diversity of *N. gonorrhoeae* MLSTs in Queensland, Australia during the COVID-19 pandemic, our study also observed a reduction in genomic diversity, as evidenced by decreased allelic diversity. Notably, our research utilised WGS data rather than amplicon sequencing[39]. Similarly, a Euro-GASP study found a decrease in genomic population diversity and increased clonality observed when comparing *N. gonorrhoeae* isolates in Europe pre and post pandemic[16]. Additionally, the decrease in gonorrhoea notifications in 2020 before rising again in 2021 could be attributed, in part, to a reduction in casual sexual encounters and potential delays in screening efforts during the pandemic. Despite the increase in notifications in 2021, there was no corresponding increase in genomic diversity observed, mirroring findings from research conducted in the Netherlands showing a sustained decrease and change in the genotypes of *N. gonorrhoeae* circulating locally[40].

The observed decrease in the effective reproductive number ($R_e$) of *N. gonorrhoeae* clusters following the implementation of social distancing measures provides empirical evidence of the effectiveness of disease control by decreasing opportunities for transmission. Notably, the emergence of new genomic clusters through the period of this study suggests that while social distancing measures may temporarily disrupt traditional patterns of transmission, new patterns of transmission dynamics may readily emerge and challenge control efforts. Further longitudinal studies may assess whether a return to pre-pandemic social and travel patterns restores *N. gonorrhoeae* genomic diversity.

In the context of our findings, there are several key areas for future research and public health action. In particular, our study highlights the critical need for ongoing genomic and epidemiological surveillance to provide granular insights into the factors promoting the emergence and spread of STIs. Second, the persistence of large genomic clusters over many years further emphasises the importance of developing novel diagnostic, treatment, and prevention strategies for gonorrhoea, including vaccines, to interrupt longitudinal, occult transmission of gonorrhoea.

## Methods

### Setting, patients and data sources
In Australia, all gonorrhoea cases are notified to public health authorities in each State or Territory. All *N. gonorrhoeae* isolated in Victoria from diagnostic laboratories were forwarded to the state Microbiological Diagnostic Unit Public Health Laboratory (MDU PHL) for AMR surveillance purposes. Between 1 January 2017 to 31 December 2017 and 1 July 2019 to 30 June 2021 (inclusive), all forwarded isolates underwent antimicrobial susceptibility testing by agar dilution and WGS. Between 1st January 2018 and 30th June 2019 (inclusive) routine WGS was not conducted, and no isolates were sequenced. Information on gonorrhoea notifications in Victoria was obtained from the National Notifiable Diseases Surveillance System (NNDSS)[1].

Samples included those collected from asymptomatic and symptomatic adult patients (≥16 years), and from genital and extra-genital (oropharyngeal and rectal) sites. Where multiple *N. gonorrheoae* genomes were isolated from the same individual on the same visit, one of these isolates was randomly selected for use in downstream transmission analyses to avoid conflating transmission cluster size. Where available, demographic information was obtained from computer-assisted self-interview records from Melbourne Sexual Health Centre (MSHC). Data collected included gender, age, site of infection, and sexual risk group (e.g. gay, bisexual, or other men who have sex with men (GBMSM), heterosexual males or females).

### Phenotypic testing and DNA extraction
Single colonies of presumptive *N. gonorrhoeae* were selected from the primary culture plate for identification and antimicrobial susceptibility testing. Isolates were confirmed as *N. gonorrhoeae* on the MALDI Biotyper (Bruker Daltonik, Bremen, Germany). Antimicrobial susceptibility testing was performed in accordance with the Australian Gonococcal Surveillance Programme (AGSP) using agar breakpoint dilution on GC agar for azithromycin, ceftriaxone, ciprofloxacin, penicillin, spectinomycin, and tetracycline. Resistance to antimicrobial agents was defined as in Supplementary Table 1. Genomic DNA was extracted from a single colony using a QIAsymphony™ DSP DNA Mini Kit (Qiagen) according to manufacturer's instructions, and sequence libraries were prepared using the NexteraXT DNA library preparation kit with accompanying indexes and 300 cycle chemistry. WGS was performed on an Illumina NextSeq 500 or 550 instrument with 150 bp paired-end reads using Illumina libraries and protocols (Illumina, San Diego, CA, USA). Reads were trimmed to remove adaptor sequences and low-quality bases (Q < 10) with Trimmomatic (v0.38)[41].

### Quality control and genome assembly
All sequences underwent quality control before further analysis. Sequences were aligned to the NCCP11945 reference genome (GenBank accession NC_011035.1) using Minimap2 (v2.17-r941) and Samtools (v1.10, using HTSlib v1.10.2) to calculate average depth of reads to the reference genome, using an inclusion criteria of ≥30x average depth[42–44]. Kraken2 (v2.1.1) was used to investigate for species contamination[45]. De novo genome assemblies were generated using Shovill (v1.1.0, https://github.com/tseemann/shovill) using default settings.

### Genotyping and genotypic antimicrobial resistance determinants
MLSTs were identified using mlst (v2.23.0; https://github.com/tseemann/mlst; PubMLST database accessed May 5, 2023) based on the alleles at seven housekeeping genes: *abcZ, adk, aroE, fumC, gdh, pdhC,* and *pgm*. Additionally, the NG-STAR typing scheme was used to determine the phenotypic profile of seven resistance genes (*penA, mtrR, porB, ponA, gyrA, parC* and *23S* rRNA), using pyngSTar ((https://github.com/leosanbu/pyngSTar, database version 2.0). NG-STAR

alleles and profiles obtained from the database hosted by the Public Health Agency of Canada (https://ngstar.canada.ca/alleles/loci_selection?lang=en)[46]. To characterise genotypic markers associated with AMR, we defined *Neisseria gonorrhoeae* Sequence Typing for Antimicrobial Resistance (NG-STAR) profiles for the dataset.

## cgMLST transmission clustering

The PubMLST *N. gonorrhoeae* cgMLST scheme consisting of 1649 genes, was prepared using chewBBACA (v2.8.5) using the PrepExternalSchema module, after which 1648 genes were accepted[26,47]. Allele calling was performed on *N. gonorrhoeae* assemblies using chewBBACA, with a training file generated by Prodigal (v2.6.3) using the reference genome NCCP11945. A relevant subset of core genes for this dataset was then determined from the schema (see Results).

A core genome multiple sequence alignment (cg-MSA) derived from cgMLST alleles for each isolate was constructed The gene sequences were retrieved based on allele number in the sequence definition database and individual gene sequences were aligned with MAFFT (v7.475), where missing alleles were converted into gaps, and the multiple sequence alignments were concatenated[48]. A 95% SNP core alignment was made from the cg-MSA using trimAl (v1.4.rev15)[49]. The resulting alignment consisted of 69,502 SNPs and was used to infer a maximum likelihood (ML) phylogenetic tree using IQ-tree (v2.0.3), with the best-fitting nucleotide substitution model chosen based on the lowest Bayesian Information Criterion (BIC)[50]. Molecular dating of ancestral events was performed using the least-squares dating (LSD) software v0.3[51].

Similar to our previous approach of using case contacts to calibrate genomic thresholds for assessing transmission the maximum pairwise allelic difference[7] for paired couples and within-site isolates was used as the threshold for subsequent single linkage hierarchical clustering when defining genomic clusters[25]. Before applying the threshold, we correct clustering for continuing *N. gonorrhoeae* evolution across our dataset, and the gap in our sampling timeframe between January 1st 2018 and June 30th 2019 by applying an inverse weighting to the pairwise allelic difference between any two isolates to account for time between sample collection according to the following formula[52]:

$$
\begin{aligned}
Corrected\ difference = &\ pairwise\ allelic\ difference \\
&- (rate\ of\ molecular\ dating\ regression \\
&\times pairwise\ temporal\ distance)
\end{aligned}
$$

## Bayesian hierarchical model

The 3,714 *N. gonorrhoeae* genomes collected after 1st July 2019 were aligned to the NCCP11945 reference genome (GenBank accession NC_011035.1) using Snippy (v4.3.5, https://github.com/tseemann/snippy), requiring a minimum of ten supporting reads and a variant frequency of 0.9 or greater. Using the pseudo-sequence generated by Snippy for each isolate, whole genome alignments were generated for each of the previously defined cgMLST clusters with at least 5 isolates. For each of these, the number of constant sites was calculated using snp-sites (v1)[53]. Following recombination filtering using Gubbins (v2.4.1) with default settings for all clusters, and a SNP alignment was generated using snp-sites, and ML phylogenetic trees were inferred using IQ-tree (v2.0.3), with the best-fitting nucleotide substitution model chosen based on the lowest BIC and the number of constant sites specified[54]. Molecular dating of ancestral events was performed on the resulting ML trees, using the least-squares dating (LSD) software (v0.3) with a rate of $4.5 \times 10^{-6}$ substitutions per site as previously defined and in line with other studies (Supplementary Fig. 15, Supplementary Table 3)[55–57]. The subsequent timed trees were used as input for a Bayesian hierarchical model.

In our Bayesian analysis, we applied a birth-death skyline process to all trees. Under this model, the epidemiological process follows a birth-death process, where branching events in a phylogenetic tree are informative about transmission, and the sampling process is directly modelled. The epidemiological parameters can change in a piecewise fashion over two intervals, with the interval time estimated as part of the model[58]. Each transmission cluster had a fixed average infection duration of three months and shared a sampling proportion parameter, meaning that sampling intensity is a free parameter that is estimated here. However, each cluster was allowed to have independent effective reproductive numbers ($Re$) and epidemic origin times. The $Re$ values were permitted to vary at a specific point in time, referred to as the "time slice," which represented a significant point of change in $Re$ values. Although $Re$ values were independent for each cluster, they were governed by a single Gamma prior distribution with two hyperparameters: shape and mean. The shape of this distribution influenced the skew, with smaller values indicating more variation in $Re$ among clusters, and larger values suggesting similarity in $Re$ values across clusters.

In our model, the $Re$ values for clusters followed a Gamma distribution, where the shape parameter reflected heterogeneity in the spread among clusters. We maintained a fixed infection duration of three months in our hierarchical model, reflecting what we considered was the most plausible scenario. The birth-death model design resulted in different magnitudes of $Re$ values, although the trends remained consistent. We assumed a uniform prior distribution for all origin times, ranging from zero to six months. Before the first genome of each cluster was sampled, we set the sampling proportion to 0.0, and thereafter, it was modelled with a Beta[1,30] prior distribution to capture our assumption that the sampling proportion was at most 10%. The hyperparameters of the Gamma distribution for $Re$ values were assigned Gamma[1,10] priors. Additionally, the time slice allowing Re value changes was assigned a uniform prior distribution between March 20 and March 30, 2020.

We sampled the posterior distribution using Markov chain Monte Carlo, implemented in BEAST2.6, with a chain length of $10^9$ steps and sampling every $10^5$ steps[59]. As the phylogenetic trees were fixed, there were no calculations of phylogenetic likelihood, making this analysis computationally more efficient compared to those involving both phylogenetic and phylodynamic likelihoods[60].

## Statistical analyses and data visualisation

Large genomic clusters ($\geq 30$) were characterised as "persistent" if the time between the earliest and latest sample collection dates was > 2 years (104 weeks), bridging the gap in sampling from 1st January 2018 to 30st June 2019. To assess variables associated with cluster persistence, we applied a multivariable logistic regression model via generalised estimating equation (GEE) to calculate adjusted odds ratios using an independence model with geepack (v1.3.9). The following specified variables were included in the models: age group, sex, size of transmission cluster, phenotypic resistance to penicillin, phenotypic resistance to tetracycline, phenotypic resistance to ciprofloxacin, phenotypic resistance or decreased susceptibility to ceftriaxone and phenotypic resistance to azithromycin. For sex, an 'unknown' category was included to accommodate missing data, with all other categories having complete data. For antimicrobial resistance phenotype, isolates were grouped binarily as either resistant or susceptible/less susceptible/decreased susceptibility except for ceftriaxone where there were no resistant persistent isolates and isolates were grouped as either decreased susceptibility or susceptible. The Mann–Whitney Rank sum test was used to compare non-normal distributions. All statistical analyses were performed in Rstudio (v2022.12.0 + 353) using base R (v4.2.2). All figures were made using ggplot2 (v3.4.0)[61].

## Reporting summary

Further information on research design is available in the Nature Portfolio Reporting Summary linked to this article.

## Data availability
Reads generated in this study and included in analyses are deposited in the NCBI database under BioProjects PRJNA520805 or PRJNA1033374 (accessions available in Supplementary Data 1). All sequencing reads from samples collected in 2017 are available in NCBI under BioProject PRJNA520805[25].

## Code availability
All code for genomic, statistical and supplementary analyses can be found in the accompanying GitHub: https://github.com/mtaouk/Neisseria_gonorrhoeae_transmission_Australia, https://doi.org/10.5281/zenodo.13316964.

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

## Acknowledgements
MLT is supported by an Australian Government Research Training Programme Scholarship. DAW is supported by a National Health and Medical Research Council (NHMRC) Investigator Grant (GNT1174555) and a Medical Research Future Fund (MRFF) Grant (FSPGN000045). BPH is supported by NHMRC Investigator Grant (GNT1196103). CKF is supported by an NHMRC Investigator Grant (APP1172900). EPFC is supported by an NHMRC Investigator Grant (GNT1172873). SD is supported by the Australian Research Council (ARC) Grant (DE190100805) and an NHMRC Investigator Grant (GNT1157586) and by the Inception Programme Investissement d' Avenir Grant (ANR-16-CONV-0005). DJI is supported by an NHMRC Investigator Grant (GNT1195210). This work was also supported by an ARC Industrial Transformation Research Hub Grant (IH190100021). MDU PHL is funded by the Victorian Government, Australia.

## Author contributions
M.L.T., G.T. and D.A.W. designed the study. D.L. performed culturing, phenotypic antimicrobial susceptibility testing and WGS. N.H., A.L., E.P.F.C., M.C. and C.K.F. provided patient metadata. M.L.T., G.T., S.J.L., C.K.H. and S.D. performed bioinformatic analysis. M.L.T., G.T., E.P.F.C. and C.K.F. performed statistical analyses. M.L.T., G.T. and S.D. prepared the figures. D.A.W., B.P.H., G.T., S.P., D.J.I. and S.J.L. provided supervision and support. The first draft of the manuscript was written by M.L.T. and D.A.W., which was revised by all authors. All authors had full access to all study data. The corresponding author had final responsibility for the decision to submit for publication.

## Competing interests
The authors declare no competing interests.

## Ethics
No individual patient consent was required or sought as data were collected in accordance with the Victorian Public Health and Wellbeing Act 2008. Ethical approval was obtained from the Alfred Hospital Ethics Committee (Project 625/17) for linking patient epidemiological data.
