## [Peer Review File · Nature Communications]

Longitudinal genomic analysis of *Neisseria gonorrhoeae* transmission dynamics in AustraliaREVIEWER COMMENTS

Reviewer #1 (Remarks to the Author):

In this manuscript, Taouk and colleagues used a longitudinal dataset of *Neisseria gonorrhoeae* whole genome sequencing data to assess the impact of antimicrobial susceptibility and demographic factors on the persistence of *N. gonorrhoeae* strains over time prior to and during COVID-19 related lockdown measures. Similar to other regions, they identify a reduction in *N. gonorrhoeae* genomic diversity during 2020. Importantly, they identified persistently transmitting clusters. These clusters were not associated with increased antimicrobial resistance but instead were associated with transmission among heterosexual populations, which they suggest may be the result of limited access to healthcare for women in disadvantaged populations. This is a unique study because of the combination of the depth of sampling in this region (~19% of all gonorrhoea infections were sequenced) and the longitudinal sampling (2017-2021). However, I have a few concerns that if addressed would strengthen the paper:

1. The references for the methods section do not seem to match with the reference list included in the manuscript. This made assessing the methods difficult, particularly when the manuscript refers to previously described approaches or results (e.g. corrected clustering, substitution rate). Additionally, references for some methods were not included (e.g. IQ-Tree).

2. Much of the analysis depends on molecular dating of phylogenies. However, the authors did not mask recombination events from alignments used for this analysis, which may inflate branch lengths and impact rates/TMRCA. Instead, the authors excluded clusters with long branch lengths under the assumption that these are impacted by recombination. It is not clear to me why only these trees could be impacted by recombination events. There are well established methods for correcting microbial phylogenies based on recombination (Gubbins, ClonalFrameML). Is there a reason these methods were not used for this analysis? Additionally, the strength of temporal signal in this data and uncertainty for estimates and phylogenies are not reported.

3. In the analysis of associations between persistence and antimicrobial susceptibility, azithromycin susceptibility was associated resistance, primarily due to two resistant lineages that did not persist; however, resistance to other antimicrobials was not associated with resistance. The authors grouped isolates with intermediate MICs are grouped with susceptible isolates rather than resistant isolates. Would the results have been different if intermediate MICs were grouped with the resistant isolates instead? Also, if the authors have access to MIC data, would it be possible to use a semi-quantitative measure for this analysis? Perhaps MICs that don't meet the resistance threshold are still contributing to persistence.

4. The authors noted a loss of diversity associated with pandemic-related lockdown. Were there any demographic or susceptibility factors associated with persistence through lockdown? I may be mistaken, but I don't think this persistence would have been captured in the reported analysis because the sampling time immediately prior to and during the pandemic is <2 years.

5. If available, I would encourage the authors to share more detailed metadata associated with the isolates in this study to enhance reproducibility. The supplementary table contains date and site of collection as well as categorical susceptibility information (but not MICs) for each antibiotic. However, analysis presented in this manuscript also depend on gender and sexual behavior, which is not reported. Additionally, if there are scripts associated with the analyses in this manuscript, they should be made publicly available as well.

Minor comments:

1. Lines 170-174: Are recombinant regions included in the alignment used to generate the phylogenetic tree and for molecular dating analysis?
2. Lines 187-188: Does this corrected distance incorporate uncertainty from the molecular dating regression? (e.g. similar to the approach in De Silva, et al. 2016)
3. Lines 217-222: Much of this information is repeated from the prior paragraph
4. Line 368: How was temporal signal assessed?
5. Lines 395-401: It's not clear to me how this method addresses the limitation of relying on a

core genome alignment since the method is based on cgMLST and within-cluster analyses are based on alignment of reads to a common reference genome.

6. Lines 451-452: Recent analysis of genomic surveillance from Europe also demonstrated a loss of diversity during 2020: [https://doi.org/10.1016/S2666-5247\(23\)00370-1](https://doi.org/10.1016/S2666-5247(23)00370-1)

7. Figure 4: I think it would be easier to see the difference between pre-March 2020 and post-March 2020 diversity if panels A and B were combined into a single figure since colors of the barplots and boxplots are already indicating pre-pandemic vs. pandemic isolates.

Reviewer #2 (Remarks to the Author):

It's been a pleasure reading your manuscript. This study provides a comprehensive analysis of *Neisseria gonorrhoeae* transmission dynamics in Australia, revealing insights into the impact of COVID-19 restrictions on gonorrhoea infections. The integration of genomic and epidemiological data highlights persistent transmission clusters and reduced genomic diversity. While the manuscript is geographically specific to Australia, it adds unique value by providing an in-depth analysis of transmission dynamics and antimicrobial resistance (AMR) patterns in the Australian context. Below is few comments I would like the authors to address.

Minor comments,

- Row 53 – Please include reference(s) to support the statement on the severe sequelae caused by gonorrhoea.

- Row 67 – Please include a reference to the recent study by Golparian et al. (Golparian D, Cole MJ, Sánchez-Busó L, Day M, Jacobsson S, Uthayakumaran T, Abad R, Bercot B, Caugant DA, Heuer D, Jansen K, Pleininger S, Stefanelli P, Aanensen DM, Bluemel B, Unemo M; Euro-GASP study group. Antimicrobial-resistant *Neisseria gonorrhoeae* in Europe in 2020 compared with in 2013 and 2018: a retrospective genomic surveillance study. *Lancet Microbe*. 2024 Apr 10:S2666-5247(23)00370-1), which discusses the impact of COVID-19 on gonorrhoea transmission in Europe, aligning with the subject of this manuscript. This study provides insights into the decreased diversity within the gonococcal population during COVID-19 lockdowns and will add depth to the introduction and discussion.

- Row 87, remove the punctuation.

Methods

- Row 129-130 – Specify which Illumina kit was used for sequencing. Please include details of the library preparation, index kit, and sequencing kit.

- Row 132. Clarify whether the reads were subjected to quality control for Q-scores or trimmed. If so, provide details about the average length of reads post-QC.

- The selected average depth of $\geq 30x$ is relatively low compared to other studies. Discuss potential impacts of regions in the gonococcal genome having low coverage ($< 10x$). Consider including additional figures in the supplementary materials to detail this aspect. Also, address how the presence of plasmids was managed, as they can cause biases in sequence depth.

- Please include how many isolates that were excluded due to quality thresholds set by the authors.

- The NG-MAST database is no longer maintained. The authors should clearly state that the NG-MAST database is v2, mainly because the NG-MAST and NG-MAST v2 are not directly comparable because the second version is slightly different in characterizing the gene sequences.

- Row 150 – Please include reference for pyngoST (aka pyngStar), Sánchez-Busó L, Sánchez-Serrano A, Golparian D, Unemo M. pyngoST: fast, simultaneous and accurate multiple sequence typing of *Neisseria gonorrhoeae* genome collections. *Microb Genom*. 2024 Jan;10(1):001189. doi: 10.1099/mgen.0.001189. PMID: 38288762; PMCID: PMC10868605.

- cgMLST. What genomes were used to define the initial cgMLST scheme? How does this scheme compare to the PubMLST cgMLST scheme for *N. gonorrhoeae*?

- cgMLST. Was the *penA* gene or 23S rRNA part of the final cgMLST? Are all the known antimicrobial resistance (AMR) genes included in the cgMLST? If not, this should be discussed though several studies have shown that antimicrobial use and the adaptation of the organism to newly introduced antibiotics drives the evolution of the species and the lack of AMR determinants might affect the phylogeny.

- cgMLST. The authors point out 5,881 cgMLST profiles was constructed. This number is confusing to read in the methods when there haven't been any mention of number of samples, cultures or patients. I would suggest that the authors mention something about the dataset in the methods before this section.
- cgMLST. The authors should include more regarding the cgMLST profiles for the isolates. How many lacked profile designation/calling? Was there a threshold to exclude isolates that lacked a certain proportion of allele designation?
- Throughout the Methods section there are parenthesis with numbers, resembling references. But I can't imagine that it is correct, are they typos? Eg Row 147 ... (6, 7) and row 160 ... (9, 10) etc. Please clarify.
- Row 181. Again, the authors refer to a gap in sampling, and the wording implies ("the gap") that the reader already know this. But the reader only knows about the gap between 1st Jan 2018 and 30th June 2019, not up to 2021. Please clarify.
- Row 193 – The authors write that snippy was used for mapping the reads to a reference genome. And for this analysis =>10x was used. How does this relate to the =>30x criteria? Is the 10x for variants? Please clarify
- Row 206. Was other models tested before the authors chose birth-death skyline?
- Row 206. Please include the reference for the birth-death skyline model. Stadler T, Kühnert D, Bonhoeffer S, Drummond AJ. Birth-death skyline plot reveals temporal changes of epidemic spread in HIV and hepatitis C virus (HCV). Proc Natl Acad Sci U S A. 2013 Jan 2;110(1):228-33. doi: 10.1073/pnas.1207965110. Epub 2012 Dec 17. PMID: 23248286; PMCID: PMC3538216.
- Row 206. Provide a brief explanation of the birth-death skyline model for clarity, especially since it's a relatively recent and complex model..
- Row 203-204 – Include the tool used for the Bayesian hierarchical analysis with version and reference.

Results

- The authors should double-check all proportions, I've just checked a few and found errors. The errors are in Table 1 as well.
 - Row 276 – please include the MIC range for ceftriaxone in the end of this sentence. A MIC of 0.25 is considered resistant according to other institutions such as CLSI and EU-CAST.
 - Row 292-293 - Please revise the sentence.
 - Row 293-294 – Please include why three isolates couldn't be assigned an NG-STAR profile or MLST (row 335).
 - Row 297 – 311 – In the first section of the Results, the authors write that it is 5881 genomes passed the QC representing one isolate from each individual. However, in this section, the authors write that pairs of isolates were grouped as four groups whereas some are from the same individual. Please clarify, is this a different dataset used for this definition?
 - In Supplementary Figure 10, please include the gene names/IDs for clarity. The multiple sequence alignment displayed is difficult to read; consider resizing or changing its orientation to landscape for better legibility. Additionally, it seems misplaced in the text around row 318. Please also provide an interpretation of the ancestral reconstructed phylogeny and detail any preprocessing of the input alignment, such as the removal of poorly aligned regions, which could influence the phylogenetic outcomes.
- Please provide a detailed interpretation of the ancestral reconstructed phylogeny to help readers understand the evolutionary implications of your findings. Additionally, clarify whether the input alignment was preprocessed to remove poorly aligned regions or other artifacts, as such modifications can significantly impact the phylogenetic conclusions.

Discussion

- Row 401 – please remove "core" before "cgMLST".
- Given the challenges associated with working with *Neisseria gonorrhoeae* that vary by geographical region, it would be beneficial if the authors could discuss how the thresholds set for the adapted version of cgMLST might perform when applied to other datasets. Can you provide insights or preliminary analysis on the adaptability and reliability of these thresholds across different geographic isolates? This discussion would greatly enhance the generalizability of your findings

Major comments

- Please discuss the overlapping findings of recently published studies highlighting any differences

in transmission dynamics pre- and post-COVID-19 specific to the Australian context or any novel antimicrobial resistance (AMR) patterns observed. This comparison is crucial for distinguishing the unique contributions of your study from existing research.

- The birth-death skyline model used in your analysis assumes constant sampling rates within each time interval. Could you clarify how this assumption was addressed, especially given potential variations in sampling effort over the study period? Additionally, it would be beneficial to know if any validation analyses were conducted to investigate the temporal signal within your dataset before performing the temporal phylogenies.

Reviewer #3 (Remarks to the Author):

This manuscript by Taouk et al. provides a longitudinal genomic analysis of *Neisseria gonorrhoeae* isolates over a four and a half year time period in Victoria, Australia. Notably, a major goal of the study was to identify transmission networks in the dataset and understand how the COVID-19 pandemic and social distancing affected gonorrhea transmission and genomic diversity. The authors were able to identify genomic clusters of isolates and how they persisted over the studied time period, with some large clusters persisting through the pandemic and social distancing periods. Clusters that persisted for longer periods of time were associated with higher ratios of women to men and susceptibility to azithromycin. The study also demonstrates the decrease in genomic diversity among the isolates following times of social distancing.

The authors have produced a strong genomic surveillance study that has the potential for a significant overall impact. There are some areas where the analyses could be more robust to enhance the interpretation of the data.

Comments and suggestions:

-Lines 136-139: Why was such a low cut-off (70%) for species composition used?

-Were any isolates associated with possible treatment failure cases?

-For international readers, it would help to add what the treatment recommendations were during the studied time period.

-Lines 285 and 293: Why couldn't NG-MAST or NG-STAR be assigned in these cases?

-Both "genomic clusters" and "transmission clusters" seem to be used interchangeably. I suggest sticking with one term. Using transmission clusters may be making some assumptions about some of the clusters where perhaps there are not sufficient epi links to infer transmission.

-Lines 434-435: Any thoughts on why persistence would be associated with azithromycin susceptibility (especially if azithromycin is being used as a treatment)?

-Lines 558-559: In the Figure 4 legend it says "unique isolates for each gene", however in the figure and results it says "unique alleles per gene", so it may be a typing error where "isolates" needs to be changed to "alleles".

-Supplementary Figure 7: When multiple isolates from a patient or multiple colonies were sequenced for an isolate, how were these chosen? Are they from different anatomic sites? Did this practice occur on a convenience basis or were they systematically chosen?

-Supplementary Figure 10: Was this type of SNP alignment done for all of the clusters?

Here are comments and suggestions specific to the phylogenomic and phylodynamic methods:

18% of the reported cases in Victoria were sequenced, which is a very good representative sample to perform population transmission dynamic analysis and infer trends especially before and after COVID-19 pandemic. The dataset is unique and most of analysis are performed appropriately. One major concern I have is that authors have not systematically removed the recombination events while performing their phylogenomic and Phylodynamic analysis. I have made a few suggestions to improve the analysis results and made some suggestions to perform some additional comparative analysis to assess whether different methods provide similar trends.

-Lines 165 – 174: Authors concatenated all the cgMLST gene (n=1648 genes) alignments and generated a whole genome core-SNP alignment, which was used for the ML phylogenetic analysis. This inferred ML tree was used for molecular dating. A root to tip correlation is not shown but it is required for timed phylogenetic analysis. I see a regression plot in Figure 1C, which appears to have a positive correlation, but it would be great to provide the estimated correlation. Considering the amount of recombination events in GC, it would be beneficial to the readers to see whether the branch lengths as well as molecular dating of ancestral events will be similar to cgMLST phylogenetic analysis if the whole genome alignment with 5881 genomes was generated using snippy followed by recombination detection using gubbins and use the resultant SNP alignment (outside of the recombinant regions) for ML phylogenetic analysis with ascertainment bias correction. The ascertainment bias correction imposed due to the usage of a SNP alignment with no recombination events will alleviate authors concerns that the use of a strict core genome alignment potentially results in isolates being classified as more closely related than they would be (as mentioned by authors in the discussion section (lines 396 – 399). I totally understand that it is inappropriate to do recombination detection and masking using gubbins on a concatenated core-gene alignment. A ML phylogenetic tree figure with all the major MLSTs mapped to it along with some important AMR markers will be a good addition to the manuscript. It would be great if the authors could also provide the SNP differences from the above suggested recombination removed SNP alignment inferred from gubbins and provide a comparison to the reported allele differences between individuals, within individuals, within sites and in paired couples (Lines 309 – 311). Also, a comparison of the estimated allelic distance per week to that of the SNPs distance per week (lines 313 to 315) will be very valuable to the entire GC genomics community.

-Lines 196 – 204: Glad to see that the authors generated separate ML trees for their clusters using snippy generated whole genome alignment, which is highly appreciated but the way of arbitrarily removing recombination regions based on the longest branch lengths is not acceptable. I would strongly recommend implementing recombination correction using gubbins on those trees displaced (supplementary figure 15) so that mostly all the clusters will be included in the analysis. Gubbins will be very efficient in recombination detection within an alignment of genetically similar isolates. Please update the timed trees as well (supplementary figure 16). It would be great if the top MLST sequence type is indicated for each of the cluster trees in supplementary figures 15 and 16). There is no way to correlate cluster designations mentioned in figure 3 to the cluster names in supplementary figures 15 and 16.

-Lines 206 – 233. It would be great if the authors could implement TransPhylo (Didelot et al. 2017) and estimate effective reproductive numbers and check whether the estimates are similar. Transphylo has been implemented previously in GC (Joseph et al. 2022; Osnes et al. 2020). Transpylo will also estimate the potential unsampled isolates in the transmission chain (see Figure S5 in (Joseph et al. 2022)) and it will be ideal to run it on the timed phylogenies estimated for the clusters.

Authors could also estimate the population trajectories estimated by modeling the effective population size over time for each of their clusters using the currently implemented birth-death skyline process similar to shown in (Joseph et al. 2022 Figure S2). This will clearly indicate the population trajectories of these clusters before, during COVID 19 restrictions and after the removal of COVID 19 restrictions.

-Didelot, Xavier, Christophe Fraser, Jennifer Gardy, and Caroline Colijn. 2017. "Genomic Infectious Disease Epidemiology in Partially Sampled and Ongoing Outbreaks." *Molecular Biology and Evolution* 34 (4): 997–1007.

-Joseph, Sandeep J., Jesse C. Thomas, Matthew W. Schmerer, John C. Cartee, Sancta St Cyr, Karen

Schlanger, Ellen N. Kersh, Brian H. Raphael, Kim M. Gernert, and Antimicrobial Resistant *Neisseria gonorrhoeae* Working Group. 2022. "Global Emergence and Dissemination of *Neisseria Gonorrhoeae* ST-9363 Isolates with Reduced Susceptibility to Azithromycin." *Genome Biology and Evolution* 14 (1). <https://doi.org/10.1093/gbe/evab287>.

-Osnes, Magnus N., Xavier Didelot, Jolinda de Korne-Elenbaas, Kristian Alfsnes, Ola B. Brynildsrud, Gaute Syversen, Øivind Jul Nilsen, Birgitte Freiesleben De Blasio, Dominique A. Caugant, and Vegard Eldholm. 2020. "Sudden Emergence of a *Neisseria Gonorrhoeae* Clade with Reduced Susceptibility to Extended-Spectrum Cephalosporins, Norway," November, mgen000480.

REVIEWER COMMENTS

REVIEWER #1

Remarks to the authors:

In this manuscript, Taouk and colleagues used a longitudinal dataset of *Neisseria gonorrhoeae* whole genome sequencing data to assess the impact of antimicrobial susceptibility and demographic factors on the persistence of *N. gonorrhoeae* strains over time prior to and during COVID-19 related lockdown measures. Similar to other regions, they identify a reduction in *N. gonorrhoeae* genomic diversity during 2020. Importantly, they identified persistently transmitting clusters. These clusters were not associated with increased antimicrobial resistance but instead were associated with transmission among heterosexual populations, which they suggest may be the result of limited access to healthcare for women in disadvantaged populations. This is a unique study because of the combination of the depth of sampling in this region (~19% of all gonorrhea infections were sequenced) and the longitudinal sampling (2017-2021). However, I have a few concerns that if addressed would strengthen the paper:

We thank the Reviewer for these positive comments.

Major comments:

1. The references for the methods section do not seem to match with the reference list included in the manuscript. This made assessing the methods difficult, particularly when the manuscript refers to previously described approaches or results (e.g. corrected clustering, substitution rate). Additionally, references for some methods were not included (e.g. IQ-Tree).

We apologise for this formatting issue. All references in the methods have now been corrected. The correct citation for each sentence is contained in the parenthesis and the corresponding full reference is available in the reference list at the end of the document.

2. Much of the analysis depends on molecular dating of phylogenies. However, the authors did not mask recombination events from alignments used for this analysis, which may inflate branch lengths and impact rates/TMRCA. Instead, the authors excluded clusters with long branch lengths under the assumption that these are impacted by recombination. It is not clear to me why only these trees could be impacted by recombination events. There are well established methods for correcting microbial phylogenies based on recombination (Gubbins, ClonalFrameML). Is there a reason these methods were not used for this analysis? Additionally, the strength of temporal signal in this data and uncertainty for estimates and phylogenies are not reported.

We thank the reviewer for raising this point. To address the concerns raised in this comment as well as by Reviewer 3 and the Editor, we have extensively reworked the analyses pertaining to the calculation of R_e from clusters, including regenerating all alignments, masking for recombination using Gubbins and regenerating phylogenetic trees, and using these new trees as input for the Bayesian analysis. Overall, we found that using Gubbins did not change the tree topology significantly, except in those clusters where a high proportion of recombination was identified – and these clusters were excluded in our original analysis. After using Gubbins we were able to include all 97 clusters in the analysis.

Specifically, we have edited the methods section to reflect incorporating Gubbins into our analysis in lines 214 – 230:

*“For each of these, the number of constant sites was calculated using *snp-sites* (v1) (43). Following recombination filtering using *Gubbins* (v2.4.1) with default settings for all clusters, and a SNP alignment was generated using *snp-sites*, and ML phylogenetic trees were inferred using *IQ-tree* (v2.0.3), with the best-fitting nucleotide substitution model chosen based on the lowest BIC and the number of constant sites specified (44). Molecular dating of ancestral events was performed on the resulting ML trees, using the least-squares dating (*LSD*) software (v0.3) with a rate of 4.5×10^{-6} substitutions per site as previously defined and in line with other studies (Supplementary Fig. 15) (45-47). The subsequent timed trees were used as input for a Bayesian hierarchical model.”*

Following our Bayesian re-analysis using all clusters excluding sites identified as recombinogenic, we saw a small increase in the mean R_e prior to March 2020 while the mean R_e following March 2020 remained the same. We have edited the results in lines 450 – 456 to reflect this:

“For clusters with at least 5 genomes, we observed a decrease in average effective reproductive number (R_e) from late March 2020 (95% credible interval between the 20th and 24th of March) coinciding with the implementation of public health measures. The mean R_e prior to the implementation of public health measures was 1.46 (95% credible interval: 1.38 to 1.55), while following the implementation of public health measures it was 1.26 (95% credible interval: 1.13 to 1.39) (Fig. 4b-c). The posterior probability of a decrease in the mean R_e after public health measures was 0.99 (corresponding Bayes factor support of 95 or ‘decisive’ evidence).”

We have also updated Fig 4b and c and Supplementary Fig. 13 – 15 to reflect the change in methods.

Figure 1.c shows the allelic divergence estimate which shows the dataset has a temporal signal – we have also included a root-to-tip regression for the dataset in response to reviewer number 3 comment 11. We used a fixed evolutionary rate of 4.5×10^{-6} substitutions/site/year, based on studies by Golparian et al. 2020 (<https://doi.org/10.1186/s12864-020-6511-6>) who estimated 4.5×10^{-6} substitutions per site per year for the species, Joseph et al. 2022 (<https://doi.org/10.1093/gbe/evab287>) who estimate 4.752×10^{-6} substitutions per site per year (95% highest posterior density [HPD] ranging from 4.163×10^{-6} to 4.933×10^{-6}) for the ST-9363 Core-Genogroup, Sanchez-Buso et al. 2020 (<https://doi.org/10.1038/s41564-019-0501-y>) who estimate 3.74×10^{-6} substitutions per site per year (confidence interval 3.39×10^{-6} to 4.07×10^{-6}), Grad et al. 2014 ([https://doi.org/10.1016/S1473-3099\(13\)70693-5](https://doi.org/10.1016/S1473-3099(13)70693-5)) who estimate 2.5×10^{-6} substitutions per site per year and Osnes et al 2021 who estimate the mean substitution rate in the non-recombining regions of the genome was 2.2×10^{-6} per site per year (confidence interval 1.5×10^{-6} to 3.2×10^{-6}). Our fixed rate is in line with these previous studies. We chose this approach because it avoids estimating this parameter for data sets that are likely too small and thus may have insufficient signal. This method has been used repeatedly for analysing pathogens, whose molecular evolutionary rate is known ‘a priori’ (Attwood et al. 2022; <https://doi.org/10.1038/s41576-022-00483-8>).

3. In the analysis of associations between persistence and antimicrobial susceptibility, azithromycin susceptibility was associated resistance, primarily due to two resistant lineages that did not persist; however, resistance to other antimicrobials was not associated with

resistance. The authors grouped isolates with intermediate MICs are grouped with susceptible isolates rather than resistant isolates. Would the results have been different if intermediate MICs were grouped with the resistant isolates instead? Also, if the authors have access to MIC data, would it be possible to use a semi-quantitative measure for this analysis? Perhaps MICs that don't meet the resistance threshold are still contributing to persistence.

Thank you for this suggestion. To address this comment, we reran the relevant models with two changes. Firstly, we adjusted our GEE model so that isolates were grouped binarily as either phenotypically resistant/less susceptible/decreased susceptibility or susceptible. The following specified variables were also included in the models: age group, sex, size of transmission cluster, phenotypic resistance to penicillin, phenotypic resistance to tetracycline, phenotypic resistance to ciprofloxacin, phenotypic resistance or decreased susceptibility to ceftriaxone and phenotypic resistance to azithromycin. For sex, an 'unknown' category was included to accommodate missing data, with all other categories having complete data. The results are as follows:

```

Coefficients:
      Estimate Std.err Wald Pr(>|W|)
(Intercept)  -3.8487  1.8957  4.12  0.0423 *
SexM         -1.0740  0.5143  4.36  0.0368 *
SexOther/Unknown -1.8078  0.7005  6.66  0.0099 **
AgeGroup20-29 -0.3327  0.3757  0.78  0.3758
AgeGroup30-39 -0.3826  0.3364  1.29  0.2554
AgeGroup40-49 -0.5965  0.4673  1.63  0.2018
AgeGroup50-59 -0.7065  0.4866  2.11  0.1465
AgeGroup60-69 -0.0922  0.6831  0.02  0.8926
AgeGroup70-79 -0.3989  1.3806  0.08  0.7726
Size         0.0136  0.0060  5.15  0.0232 *
PENSUS      2.5868  0.9328  7.69  0.0056 **
TETSUS      0.8578  0.6361  1.82  0.1775
CTRIXSUS   -1.7719  1.1047  2.57  0.1087
CIPROSUS    1.6711  1.0114  2.73  0.0985 .
AZITHSUS    4.6184  0.9599 23.15 1.5e-06 ***
---
Signif. codes:  0 '***' 0.001 '**' 0.01 '*' 0.05 '.' 0.1 ' ' 1

Correlation structure = independence
Estimated Scale Parameters:

      Estimate Std.err
(Intercept)  0.622  0.513
Number of clusters:  31 Maximum cluster size: 709

```

```
# A tibble: 15 × 7
  term                estimate std.error statistic    p.value  conf.low conf.high
  <chr>                <dbl>    <dbl>    <dbl>    <dbl>    <dbl>    <dbl>
1 (Intercept)         0.0213    1.90      4.12  0.0423    0.000519  0.875
2 SexM                0.342     0.514     4.36  0.0368    0.125     0.936
3 SexOther/Unknown    0.164     0.700     6.66  0.00986   0.0416    0.647
4 AgeGroup20-29       0.717     0.376     0.784  0.376     0.343     1.50
5 AgeGroup30-39       0.682     0.336     1.29  0.255     0.353     1.32
6 AgeGroup40-49       0.551     0.467     1.63  0.202     0.220     1.38
7 AgeGroup50-59       0.493     0.487     2.11  0.147     0.190     1.28
8 AgeGroup60-69       0.912     0.683     0.0182 0.893     0.239     3.48
9 AgeGroup70-79       0.671     1.38      0.0835 0.773     0.0448    10.0
10 Size               1.01      0.00600   5.15  0.0232    1.00      1.03
11 PENSUS              13.3      0.933     7.69  0.00555   2.13     82.7
12 TETSUS              2.36      0.636     1.82  0.178     0.678     8.20
13 CTRIXSUS           0.170     1.10      2.57  0.109     0.0195    1.48
14 CIPROSUS            5.32      1.01      2.73  0.0985    0.733     38.6
15 AZITHSUS            101.      0.960     23.1  0.00000150 15.4     665.
```

Secondly, we adjusted our GEE model so that continuous MIC values were used for the phenotypic antimicrobial susceptibility profiles. The following specified variables were included in the models: age group, sex, size of transmission cluster, phenotypic resistance to penicillin, phenotypic resistance to tetracycline, phenotypic resistance to ciprofloxacin, phenotypic resistance or decreased susceptibility to ceftriaxone and phenotypic resistance to azithromycin. For sex, an ‘unknown’ category was included to accommodate missing data, with all other categories having complete data. The results are as follows:

```
Coefficients:
  Estimate Std.err Wald Pr(>|W|)
(Intercept) 4.928596 1.617803 9.281 0.00232 **
SexM        -1.275334 0.347233 13.490 0.00024 ***
SexOther    -2.147489 0.721981 8.847 0.00294 **
AgeGroup20-29 -0.091852 0.345161 0.071 0.79015
AgeGroup30-39 -0.258350 0.323177 0.639 0.42405
AgeGroup40-49 -0.452894 0.419387 1.166 0.28019
AgeGroup50-59 -0.468398 0.499611 0.879 0.34849
AgeGroup60-69 -0.547614 0.609648 0.807 0.36905
AgeGroup70-79 0.018856 0.699239 0.001 0.97849
Size         0.017994 0.006914 6.774 0.00925 **
PEN          -0.245739 0.376097 0.427 0.51350
TET          -0.211504 0.297985 0.504 0.47784
CTRIX       -0.056818 0.629740 0.008 0.92811
CIPRO       -0.143813 0.162365 0.785 0.37576
AZITH       -0.959885 0.353599 7.369 0.00664 **
---
Signif. codes:  0 '***' 0.001 '**' 0.01 '*' 0.05 '.' 0.1 ' ' 1
```

```
Correlation structure = independence
Estimated Scale Parameters:
```

```
      Estimate Std.err
(Intercept) 0.6458 2.57
Number of clusters: 31 Maximum cluster size: 709
```

```
# A tibble: 15 × 7
  term      estimate std.error statistic  p.value conf.low conf.high
<chr>      <dbl>      <dbl>      <dbl>    <dbl>   <dbl>   <dbl>
1 (Intercept)  138.        1.62        9.28    0.00232  5.80    3293.
2 SexM         0.279      0.347      13.5    0.000240 0.141    0.552
3 SexOther     0.117      0.722       8.85    0.00294  0.0284   0.481
4 AgeGroup20-29 0.912      0.345      0.0708  0.790    0.464    1.79
5 AgeGroup30-39 0.772      0.323      0.639    0.424    0.410    1.46
6 AgeGroup40-49 0.636      0.419      1.17     0.280    0.279    1.45
7 AgeGroup50-59 0.626      0.500      0.879    0.348    0.235    1.67
8 AgeGroup60-69 0.578      0.610      0.807    0.369    0.175    1.91
9 AgeGroup70-79 1.02       0.699      0.00727  0.978    0.259    4.01
10 Size        1.02      0.00691    6.77    0.00925  1.00    1.03
11 PEN         0.782      0.376      0.427    0.514    0.374    1.63
12 TET         0.809      0.298      0.504    0.478    0.451    1.45
13 CTRIX       0.945      0.630      0.00814  0.928    0.275    3.25
14 CIPRO       0.866      0.162      0.785    0.376    0.630    1.19
15 AZITH       0.383      0.354      7.37     0.00664  0.191    0.766
```

We see that an individual’s sex and the size of the genomic cluster are still significantly associated with persistence of clusters. The trends in phenotypic azithromycin susceptibility patterns are consistent across these two models (and our original results), although the first model here finds penicillin sensitivity as statistically associated with persistence. We chose to use the original binary model as it better reflects clinical breakpoints.

4. The authors noted a loss of diversity associated with pandemic-related lockdown. Were there any demographic or susceptibility factors associated with persistence through lockdown? I may be mistaken, but I don’t think this persistence would have been captured in the reported analysis because the sampling time immediately prior to and during the pandemic is <2 years.

The dataset includes nine months directly prior to and 15 months during the pandemic-related lockdowns. We do not present any data following comparable removal of pandemic-related public health measures, which would commence in November 2022 for our geographic region. Our study focuses on the difference in genomic diversity across our sampling timeframe rather than demographic factors.

5. If available, I would encourage the authors to share more detailed metadata associated with the isolates in this study to enhance reproducibility. The supplementary table contains date and site of collection as well as categorical susceptibility information (but not MICs) for each antibiotic. However, analysis presented in this manuscript also depend on gender and sexual behaviour, which is not reported. Additionally, if there are scripts associated with the analyses in this manuscript, they should be made publicly available as well.

We have incorporated this suggestion and included more detailed metadata in the supplementary dataset.

We have made all code and additional supplementary analyses available in a GitHub (https://github.com/mtaouk/Neisseria_gonorrhoeae_transmission_Australia) associated with this publication. This has been included in line 622 – 625 of the manuscript under the Code Availability section.

Minor comments:

6. Lines 170-174: Are recombinant regions included in the alignment used to generate the phylogenetic tree and for molecular dating analysis?

The alignment of all 5,881 isolates included in this analysis was generated by concatenating the alleles typed at each cgMLST gene for each isolate – a method described previously in Hennart et al. 2022 (<https://doi.org/10.1093/molbev/msac135>) and Samuelson et al. 2017 (<https://doi.org/10.1371/journal.pone.0187832>). It would be inappropriate to do recombination detection and masking on a concatenated cg-MSA. For example, ClonalFrameML specifies that their tool is “especially aimed at analysis of whole genome sequences” (<https://doi.org/10.1371/journal.pcbi.1004041>). Further, Gubbins would not be appropriate in this case as the authors specify that the recombination algorithm “is most effective when detecting imports of sequence into a densely sampled collection of closely-related isolates, where recombinations import a high density of base substitutions from divergent donors” (<https://doi.org/10.1093/nar/gku1196>). As this dataset represents a very widely sampled and diverse set of isolates, recombination detection tools like Gubbins would not be suitable so no recombination filtering or masking was undertaken.

7. Lines 187-188: Does this corrected distance incorporate uncertainty from the molecular dating regression? (e.g. similar to the approach in De Silva, et al. 2016)

This method was based in the approach detailed in De Silva, et al. 2016. The corrected distance does not incorporate uncertainty from the molecular dating regression as the molecular evolution of *N. gonorrhoeae* in this dataset was estimated to be 4.995×10^{-2} allelic distance per week and we provide a 95% credibility interval to support this: $4.994 \times 10^{-2} - 5.000 \times 10^{-2}$.

8. Lines 217-222: Much of this information is repeated from the prior paragraph

We appreciate this may read similar to the section detailing how we generated a whole dataset phylogeny, but this section refers to the individual cluster alignments generated using snippy. This section has now been heavily edited following addressing reviewer comments suggesting reanalysis.

9. Line 368: How was temporal signal assessed?

Please see our reply above to comment #2 from Reviewer 1. We fixed the evolutionary rate of individual clusters to a known value because their small size means that they may not have sufficient signal to estimate this parameter. For a root-to-tip regression of the entire dataset please see the response to Reviewer 3 comment 11.

10. Lines 395-401: It’s not clear to me how this method addresses the limitation of relying on a core genome alignment since the method is based on cgMLST and within-cluster analyses are based on alignment of reads to a common reference genome.

Our method addresses the limitation of relying on a core genome alignment by using cgMLST allelic differences rather than SNP distances to determine relatedness between our isolates. SNP distances can be significantly influenced by recombination events, where large regions of diversity might be mistakenly counted as multiple variants. In contrast, the cgMLST approach treats any difference within a gene as a single pairwise difference, minimizing the impact of recombination on clustering analyses. We present an example in Supplementary Fig. 11:

“A SNP alignment for genomic cluster 230 showing that 16 isolates all share the same 177 SNPs (position 224 – 401). These SNPs can all be found within 4 consecutive genes in the

genome (NGO_2109, NGO11390, NGO2111 and NGO2112). As all 16 isolates sharing SNPs in these genes were collected before June 2020, this may indicate a recombination event that occurred sometime before 2020, resulting in 177 SNPs which would be treated as 177 individual evolutionary events, instead of a single recombination by SNP based phylogenetic methods. While cgMLST summarises these 177 SNPs into 4 allelic differences, highlighting the appropriateness of the cgMLST clustering method for this dataset.”

Therefore, our method reduces the potential bias introduced by recombination, enhancing the accuracy of our clustering results.

Further we have included a new supplementary analysis which shows that a 100% strict core genome is not particularly useful in this large and diverse dataset of *N. gonorrhoeae* as the number of sites present in 100% of all isolates is extremely small (4585bp) meaning very little phylogenetic signal remains to calculate pairwise SNP distances and infer relatedness between isolates and generate phylogenies. We have detailed this in the Supplementary Appendix and in the GitHub associated with this publication.

11. Lines 451-452: Recent analysis of genomic surveillance from Europe also demonstrated a loss of diversity during 2020: [https://doi.org/10.1016/S2666-5247\(23\)00370-1](https://doi.org/10.1016/S2666-5247(23)00370-1)

We thank the reviewer for this useful reference. We have included it in lines 556 – 558:

*“Similarly, a Euro-GASP study found a decrease in genomic population diversity and increased clonality observed when comparing *N. gonorrhoeae* isolates in Europe pre and post pandemic (65)”*

12. Figure 4: I think it would be easier to see the difference between pre-March 2020 and post-March 2020 diversity if panels A and B were combined into a single figure since colours of the barplots and boxplots are already indicating pre-pandemic vs. pandemic isolates.

We agree this would improve the clarity of the plot and have combined these two panels in the figure and edited the text accordingly.

REVIEWER #2:

Remarks to the authors:

It’s been a pleasure reading your manuscript. This study provides a comprehensive analysis of *Neisseria gonorrhoeae* transmission dynamics in Australia, revealing insights into the impact of COVID-19 restrictions on gonorrhoea infections. The integration of genomic and epidemiological data highlights persistent transmission clusters and reduced genomic diversity. While the manuscript is geographically specific to Australia, it adds unique value by providing an in-depth analysis of transmission dynamics and antimicrobial resistance (AMR) patterns in the Australian context. Below are a few comments I would like the authors to address.

We thank the Reviewer for these positive comments.

Minor comments:

1. Row 53 – Please include reference(s) to support the statement on the severe sequelae caused by gonorrhoea.

The following publication has been added as a reference at line 53: Unemo M, Shafer WM. Antibiotic resistance in *Neisseria gonorrhoeae*: origin, evolution, and lessons learned for the future. *Ann N Y Acad Sci*. 2011 Aug;1230:E19-28. doi: 10.1111/j.1749-6632.2011.06215.x. PMID: 22239555; PMCID: PMC4510988.

2. Row 67 – Please include a reference to the recent study by Golparian et al. (Golparian D, Cole MJ, Sánchez-Busó L, Day M, Jacobsson S, Uthayakumaran T, Abad R, Bercot B, Caugant DA, Heuer D, Jansen K, Pleininger S, Stefanelli P, Aanensen DM, Bluemel B, Unemo M; Euro-GASP study group. Antimicrobial-resistant *Neisseria gonorrhoeae* in Europe in 2020 compared with in 2013 and 2018: a retrospective genomic surveillance study. *Lancet Microbe*. 2024 Apr 10:S2666-5247(23)00370-1), which discusses the impact of COVID-19 on gonorrhoea transmission in Europe, aligning with the subject of this manuscript. This study provides insights into the decreased diversity within the gonococcal population during COVID-19 lockdowns and will add depth to the introduction and discussion.

We thank the reviewer for this useful reference. We have included it in lines 556 – 558:

“Similarly, a Euro-GASP study found a decrease in genomic population diversity and increased clonality observed when comparing N. gonorrhoeae isolates in Europe pre and post pandemic (65)”

3. Row 87, remove the punctuation.

We thank the reviewer for finding this typo – it has now been corrected.

4. Row 129-130 – Specify which Illumina kit was used for sequencing. Please include details of the library preparation, index kit, and sequencing kit.

We have included a sentence to clarify this in line 130 – 131:

“Sequence libraries were prepared using the NexteraXT DNA library preparation kit with accompanying indexes and 300 cycle chemistry.”

5. Row 132. Clarify whether the reads were subjected to quality control for Q-scores or trimmed. If so, provide details about the average length of reads post-QC.

We have clarified this at line 133 – 134:

“Reads were trimmed to remove adaptor sequences and low-quality bases ($Q < 10$) with Trimmomatic (v0.38) (26)”

The average read length for each isolate is now listed in the Supplementary Dataset. The overall mean average read length across the dataset was 148.5 bp.

6. The selected average depth of $\geq 30x$ is relatively low compared to other studies. Discuss potential impacts of regions in the gonococcal genome having low coverage ($< 10x$). Consider including additional figures in the supplementary materials to detail this aspect. Also, address how the presence of plasmids was managed, as they can cause biases in sequence depth.

Before quality checks and filtering, the mean sequencing depth for the 6,329 isolates was 134.9x (range: 2.8x to 1,971x). We excluded 5 isolates due to low sequencing depth (<30x). For clarity, we have now included the sequencing depth for each isolate in the Supplementary Dataset and added a comprehensive plot showing QC results in Supplementary Figure 2.

To address potential impacts of low coverage regions (<10x) in the gonococcal genome, our cgMLST analysis was based on assemblies. This means that as long as we had good assemblies from the resulting reads, we could accurately define alleles for our specified genes and proceed with downstream analyses. As a measure to account for this, we only included isolates with assemblies of 300 or less contigs. For our snippy alignments used in downstream Bayesian modelling, we required a minimum depth of 10 reads at any site, with at least 90% of those reads supporting the variant for it to be included. This threshold helps mitigate the impact of any low coverage regions on our analyses.

Regarding the presence of plasmids, these were not included in our cgMLST or snippy analyses and were therefore not used in calculating the average read depth. The average read depth was calculated by counting the number of reads aligning to the NCCP11945 reference genome, multiplied by the average read length for each isolate, and divided by the NCCP11945 genome size. All read data generated or used in this study has been made available on NCBI for those interested in plasmid sequences and relative sequence coverage.

7. Please include how many isolates that were excluded due to quality thresholds set by the authors.

We have included the following in the Supplementary Appendix (line 16 – 22), with reference to it in the main text (line 309):

“Of the 6329 isolates sequenced, 8 were excluded due to an average sequencing depth of <30, and 85 were excluded due to assemblies of ≥ 300 or 0 contigs. 2 isolates were excluded as they were missing an accompanying date of collection, 2 were excluded as they were missing an accompanying age of individual, and 17 were excluded as they were collected from individuals under the age of 16. 6,215 isolates remained. Of these, 49 were duplicate colony picks and 285 were collected from the same individual on the same day at different body sites and were not included for clustering analyses. 5,881 isolates were included for clustering and downstream analyses.”

8. The NG-MAST database is no longer maintained. The authors should clearly state that the NG-MAST database is v2, mainly because the NG-MAST and NG-MAST v2 are not directly comparable because the second version is slightly different in characterizing the gene sequences.

We have clarified that the NG-MAST v2.0 database was used in the methods section (lines 148).

9. Row 150 – Please include reference for pyngoST (aka pyngSTar), Sánchez-Busó L, Sánchez-Serrano A, Golparian D, Unemo M. pyngoST: fast, simultaneous and accurate multiple sequence typing of *Neisseria gonorrhoeae* genome collections. Microb Genom. 2024 Jan;10(1):001189. doi: 10.1099/mgen.0.001189. PMID: 38288762; PMCID: PMC10868605.

This citation has been added at line 163.

10. cgMLST. What genomes were used to define the initial cgMLST scheme? How does this scheme compare to the PubMLST cgMLST scheme for *N. gonorrhoeae*?

The scheme we used here is the same as the PubMLST cgMLST scheme. We have referenced the scheme in the methods. We apologise for the ambiguity and have made this clearer in the text at line 168:

*“The PubMLST *N. gonorrhoeae* cgMLST scheme consisting of 1,649 genes, was prepared using chewBBACA (v2.8.5) using the PrepExternalSchema module, after which 1,648 genes were accepted”*

11. cgMLST. Was the *penA* gene or 23S rRNA part of the final cgMLST? Are all the known antimicrobial resistance (AMR) genes included in the cgMLST? If not, this should be discussed though several studies have shown that antimicrobial use and the adaptation of the organism to newly introduced antibiotics drives the evolution of the species and the lack of AMR determinants might affect the phylogeny.

penA was part of the final scheme however 23S rRNA was not, as it is not part of the original scheme. The original cgMLST was built using Roary, 'Roary identifies bacterial pan-genomes using WGS data annotated with Prokka. The gene_presence_absence output file from Roary was used to identify loci present in >95% of the dataset, representing loci “core” to the gonococcus' (Harrison et al. 2020, 10.1093/infdi/jiaa002). This is a gene-centric approach which will not pick up intergenic or non-coding sequences, such as ribosomal RNAs, for example, Roary works by taking individual protein-coding sequences, predefined using Prokka annotation, and assigning each to a single cluster of homologous sequences. This approach thus excludes non protein-coding intergenic regions (IGRs) that typically account for approximately 15% of the genome. The 23S RNA genes are likely present in all samples but not included in the cgMLST scheme because of the choice of pangenome tool used to generate the scheme. Resistance determinants which are present in 95% of isolates are included in the scheme. Plasmid borne resistance determinants are not included in the scheme the scheme only includes chromosomal loci.

Technically, genes under positive selection may evolve at varied, non-clock-like rates given this selective pressure. As these genes represent <0.5% of the total cgMLST scheme, the influence of this should be minimal on phylogenetic inference. In determining transmission groups, the reviewer is correct that the rapid evolution of AMR may lead to some groups being disaggregated, although the same could be said for any gene hypothetically under positive selection from the host or environment. We have opted to leave these genes in the scheme rather than attempting to mask or remove all genes hypothetically under positive selection.

Please also see the response to Reviewer 2 Comment 25.

12. cgMLST. The authors point out 5,881 cgMLST profiles was constructed. This number is confusing to read in the methods when there haven't been any mention of number of samples, cultures or patients. I would suggest that the authors mention something about the dataset in the methods before this section.

We apologise for the confusion. The 5,881 refers to the number of genomes included in the analyses and each of their respective cgMLST profiles. We agree that this value is not informative in this section and have changed the sentence for clarity at line 175:

“A core genome multiple sequence alignment (cg-MSA) derived from cgMLST alleles for each isolate was constructed.”

13. cgMLST. The authors should include more regarding the cgMLST profiles for the isolates. How many lacked profile designation/calling? Was there a threshold to exclude isolates that lacked a certain proportion of allele designation?

We do not assign an overall profile based on allele calling. We used the allele calling to generate a distance matrix and clustered isolates in that way so every isolate was given a cluster ID in this study which can be found in the Supplementary Dataset. Following the exclusion of loci which are not present in 95% of our genomes (explained in detail in the manuscript), the remaining 1,495 loci were very well typed within our dataset. Of our 6,215 isolates (including intra-patient and colony pick isolates), on average 1.9 loci were missing/non typable per isolate, with an average of 99.9% of loci being designated an allele for each isolate (range: 98.2% to 100%). As this was so high there was no exclusion criteria set and no genomes were excluded. We have included a sentence in the results to reflect this (lines 372 – 373):

“Following adaptation of the cgMLST scheme a mean of 99.9% (range: 98.2% to 100%) of loci were designated an allele for each isolate.”

14. Throughout the Methods section there are parenthesis with numbers, resembling references. But I can't imagine that it is correct, are they typos? Eg Row 147 ... (6, 7) and row 160 ... (9, 10) etc. Please clarify.

We apologise for this formatting issue. All references in the methods have now been corrected. The correct citation for each sentence is contained in the parenthesis and the corresponding full reference is available in the reference list at the end of the document.

15. Row 181. Again, the authors refer to a gap in sampling, and the wording implies (“the gap”) that the reader already know this. But the reader only knows about the gap between 1st Jan 2018 and 30th June 2019, not up to 2021. Please clarify.

To clarify we, first mention the gap in sampling in the first paragraph of the methods section – *“Between 1 January 2017 to 31 December 2017 and 1 July 2019 to 30 June 2021 (inclusive)”* (lines 105 – 106). We then reference this gap again in the line you have pointed out, however there was a typo and this has been corrected to *“the gap in our sampling timeframe between January 1st 2018 and June 30th 2019”* (line 199).

16. Row 193 – The authors write that snippy was used for mapping the reads to a reference genome. And for this analysis =>10x was used. How does this relate to the =>30x criteria? Is the 10x for variants? Please clarify

Please see the response to Reviewer 2 comment 6.

17. Row 206. Was other models tested before the authors chose birth-death skyline?

No. To the best of our knowledge there is no other model with the flexible parameterisation capabilities of the birth-death that can take advantage of the sampling times and sequence data. Moreover, model testing for Bayesian hierarchical has fallen out of favour because model selection can depend on the prior (Featherstone et al. 2023; <https://doi.org/10.1093/molbev/msad132>).

18. Row 206. Please include the reference for the birth-death skyline model. Stadler T, Kühnert D, Bonhoeffer S, Drummond AJ. Birth-death skyline plot reveals temporal changes of epidemic spread in HIV and hepatitis C virus (HCV). Proc Natl Acad Sci U S A. 2013 Jan 2;110(1):228-33. doi: 10.1073/pnas.1207965110. Epub 2012 Dec 17. PMID: 23248286; PMCID: PMC3538216.

This reference has been added to line 236.

19. Row 206. Provide a brief explanation of the birth-death skyline model for clarity, especially since it's a relatively recent and complex model..

We have included a brief explanation in the manuscript at lines 232 – 236:

“Under this model, the epidemiological process follows a birth-death process, where branching events in a phylogenetic tree are informative about transmission, and the sampling process is directly modelled. The epidemiological parameters can change in a piecewise fashion over two intervals, with the interval time estimated as part of the model (46).”

20. Row 203-204 – Include the tool used for the Bayesian hierarchical analysis with version and reference.

The tool was BEAST2.6. We have clarified this and included the reference in line 274.

21. The authors should double-check all proportions, I've just checked a few and found errors. The errors are in Table 1 as well.

We thank the reviewer for checking this. We have double checked and can confirm all values and percentages are now correct.

22. Row 276 – please include the MIC range for ceftriaxone in the end of this sentence. A MIC of 0.25 is considered resistant according to other institutions such as CLSI and EU-CAST.

Breakpoint MICs for each antimicrobial tested are presented in the Supplementary Table 1. These are according to the Australian Gonococcal Surveillance Program (AGSP) criteria and will not be directly comparable to CLSI and EU-CAST. For further transparency, the MIC for each isolate are now listed in the Supplementary Dataset.

23. Row 292-293 - Please revise the sentence.

We have changed this sentence to: *“There were 270 NG-STAR profiles defined in the dataset overall”* at line 356.

24. Row 293-294 – Please include why three isolates couldn't be assigned an NG-STAR profile or MLST (row 335).

2,483 isolates could not be assigned an NG-MAST profile as for one or both genes (*tpbB* and *porB*). This is because NG-MASTER identifies (a) a novel full-length alleles similar to a known allele, (b) a partial match to a known allele, (c) a missing allele or (d) multiple alleles. The allele results for the two genes can be found in the Supplementary Dataset.

For the 3 isolates that could not be assigned an NG-STAR profile, they were not able to be typed at *porB* gene which is required for an overall profile.

25. Row 297 – 311 – In the first section of the Results, the authors write that it is 5881 genomes passed the QC representing one isolate from each individual. However, in this section, the authors write that pairs of isolates were grouped as four groups whereas some are from the same individual. Please clarify, is this a different dataset used for this definition?

It is correct that 5,881 isolates represent a single infection, and these are the isolates used for clustering analyses. However, 6,215 isolates were sequenced including within-individual and within-site isolates. These were used for the calibration analyses but not clusters so as not to inflate cluster size. The epidemiologically linked pairs were 16 isolates within the 5,881 genome dataset. To make this clearer we have included an explanation in the first paragraph of the results (line 305 – 313):

“In total, 6,329 N. gonorrhoeae isolates underwent WGS and 6,215 (98.2%) passed quality control checks. 5,608 genomes represented unique infections collected from one individual each. The dataset also included 558 genomes collected from different body sites from 273 individuals (within-individual samples). A further 49 genomes were colony picks which were all collected from 13 individuals in total (within-site samples) (Supplementary Figure 2, Supplementary Dataset). In total, 5,881 genomes (one sample collected per individual, per day; this was randomly selected for within-individual samples) were included for clustering and epidemiological analyses making up 18.9% of all gonorrhoea notifications in Victoria for this period (Supplementary Dataset).”

26. In Supplementary Figure 10, please include the gene names/IDs for clarity. The multiple sequence alignment displayed is difficult to read; consider resizing or changing its orientation to landscape for better legibility. Additionally, it seems misplaced in the text around row 318. Please also provide an interpretation of the ancestral reconstructed phylogeny and detail any preprocessing of the input alignment, such as the removal of poorly aligned regions, which could influence the phylogenetic outcomes.

Please provide a detailed interpretation of the ancestral reconstructed phylogeny to help readers understand the evolutionary implications of your findings. Additionally, clarify whether the input alignment was preprocessed to remove poorly aligned regions or other artifacts, as such modifications can significantly impact the phylogenetic conclusions.

We have altered Supplementary Figure 11 (previously Supplementary Figure 10) to be clearer and included the loci tags for the four genes in the Supplementary Appendix lines 87 – 88.

The methods for generating the input alignment are detailed in the methods of the manuscript at lines 175 – 192:

“A core genome multiple sequence alignment (cg-MSA) derived from cgMLST alleles for each isolate was constructed. The gene sequences were retrieved based on allele number in the

sequence definition database and individual gene sequences were aligned with MAFFT (v7.475), where missing alleles were converted into gaps, and the multiple sequence alignments were concatenated (38). A 95% SNP core alignment was made from the cg-MSA using trimAl (v1.4.rev15) (39). The resulting alignment consisted of 69,502 SNPs and was used to infer a maximum likelihood (ML) phylogenetic tree using IQ-tree (v2.0.3), with the best-fitting nucleotide substitution model chosen based on the lowest Bayesian Information Criterion (BIC) (40). Molecular dating of ancestral events was performed using the least-squares dating (LSD) software v0.3 (41)”

The full code can be found on the GitHub associated with the manuscript (https://github.com/mtaouk/Neisseria_gonorrhoeae_transmission_Australia).

We have included details of the ancestral reconstructed phylogeny within the Supplementary Appendix lines 75 – 78:

“Tree rate: 2.399 x 10⁻⁵, tMRCA: 1873.125327. The phylogenetic tree illustrates the evolutionary relationships among the dataset, with each branch representing genetic divergence events.”

27. Row 401 – please remove “core” before “cgMLST”.

This has been removed.

28. Given the challenges associated with working with *Neisseria gonorrhoeae* that vary by geographical region, it would be beneficial if the authors could discuss how the thresholds set for the adapted version of cgMLST might perform when applied to other datasets. Can you provide insights or preliminary analysis on the adaptability and reliability of these thresholds across different geographic isolates? This discussion would greatly enhance the generalizability of your findings

Thank you for this suggestion. We have applied our cgMLST clustering method and threshold to 490 *N. gonorrhoeae* genomes made available by et De Silva al. 2016. These genomes have similar metadata to our study (within-individual, within-site). We applied the same cgMLST scheme, allele calling and refining methods as described in our study. After performing allele calling on the De Silva genomes, we refined the cgMLST schema to 1,506 genes present in 95% of isolates and used these genes as the schema. This was comparable to our refined scheme of 1,495 genes. 1,476 of the genes retained in the De Silva 95% schema were included in our refined schema. Here we show the overall distribution of pairwise allelic differences across the De Silva genomes:

To determine an appropriate clustering threshold for the De Silva dataset we plotted the distribution of pairwise allelic differences of the within-individual pairs and within-site pairs of genomes, applying the same principle as for selecting the clustering threshold in our own dataset. All within-site pairs of genomes were between 0 and 3 allelic differences apart. The within-individual pairs of genomes were between 0 and 944 allelic differences apart. However, the majority of within-individual pairs fell within the 0 to 7 range. This is comparable to our own results, where we also saw co-infections amongst our within-individual pairs, representing the high pairwise allelic differences. The majority of the within-individual isolates fell below 7 pairwise allelic differences suggesting that it would be an appropriate threshold to use in this case. While the datasets and scheme are not identical, the same methods can be applied to any *N. gonorrhoeae* dataset to investigate transmission and a threshold can be applied based on available calibration isolates as done here, or a similar threshold of 7 pairwise allelic differences could be generalizable to other studies if no calibration isolates are available.

When applying the threshold of 7 allelic differences as a threshold for single linkage hierarchical clustering to defining genomic transmission clusters 32 singletons ($n = 1$), 70 pairs ($n = 2$) and 46 clusters ($n \geq 3$) were identified.

We agree this is useful information and have included it in the Supplementary Appendix and in the Supplementary Analyses detailed in the GitHub associated with this publication (https://github.com/mtaouk/Neisseria_gonorrhoeae_transmission_Australia/tree/main/Supplementary_analyses/cgMLST_method).

Major comments:

29. Please discuss the overlapping findings of recently published studies highlighting any differences in transmission dynamics pre- and post-COVID-19 specific to the Australian context or any novel antimicrobial resistance (AMR) patterns observed. This comparison is crucial for distinguishing the unique contributions of your study from existing research.

We have edited the following sentences in lines 554 – 560 of the discussion to better address this comment:

*“Consistent with other findings that indicate a decline in the diversity of *N. gonorrhoeae* MLSTs in Queensland, Australia during the COVID-19 pandemic, our study also observed a reduction in genomic diversity, as evidenced by decreased allelic diversity. Notably, our research utilised WGS data rather than amplicon sequencing. (<https://pubmed.ncbi.nlm.nih.gov/37045547/>).”*

30. The birth-death skyline model used in your analysis assumes constant sampling rates within each time interval. Could you clarify how this assumption was addressed, especially given potential variations in sampling effort over the study period? Additionally, it would be

beneficial to know if any validation analyses were conducted to investigate the temporal signal within your dataset before performing the temporal phylogenies.

We assume that sampling is constant after the detection of the first case in each cluster. We clarified this point in the manuscript. Our sampling proportion parameter is linked for all clusters, such that this parameter collapses to a weighted average. The alternative approach is to estimate a single parameter for each cluster, but doing so dramatically increases the number of parameters and reduces the benefit of the hierarchical approach. Moreover, our model estimates the time when R_e changed, such that additional intervals is likely to result in non-identifiability.

Please see our response to point #2 from Reviewer 1 about temporal signal. We fixed the evolutionary rate to a known value to avoid estimating this parameter with small transmission clusters.

REVIEWER #3:

Remarks to the authors:

This manuscript by Taouk et al. provides a longitudinal genomic analysis of *Neisseria gonorrhoeae* isolates over a four and a half year time period in Victoria, Australia. Notably, a major goal of the study was to identify transmission networks in the dataset and understand how the COVID-19 pandemic and social distancing affected gonorrhoea transmission and genomic diversity. The authors were able to identify genomic clusters of isolates and how they persisted over the studied time period, with some large clusters persisting through the pandemic and social distancing periods. Clusters that persisted for longer periods of time were associated with higher ratios of women to men and susceptibility to azithromycin. The study also demonstrates the decrease in genomic diversity among the isolates following times of social distancing.

The authors have produced a strong genomic surveillance study that has the potential for a significant overall impact. There are some areas where the analyses could be more robust to enhance the interpretation of the data.

We thank the Reviewer for these positive comments.

Comments and suggestions:

1. Lines 136-139: Why was such a low cut-off (70%) for species composition used?

No isolates were excluded on the basis of Kraken percentage alone; this score was used to investigate for contamination. Overall, the mean species composition for all 6,329 isolates sequenced was 91.01 (range: 1.9 to 96.9). We selected 70% as 99.84% (6,319) of isolates had a score higher than that, and only 99.71% (6,311) isolates had a score higher than 80%. No isolates were excluded on the basis of Kraken percentage alone.

For clarity, we have now included the Kraken result for each isolate in the Supplementary Dataset, as well as including the spread of results in the Supplementary Appendix in Supplementary Figure 2. We have changed the text to “*Kraken (v2.1.1) was used to investigate for contamination*” (lines 140 – 141) to clarify our use of the tool here.

2. Were any isolates associated with possible treatment failure cases?

Unfortunately, we did not have access to this level of clinical metadata and cannot comment on this.

3. For international readers, it would help to add what the treatment recommendations were during the studied time period.

We have included a sentence in the discussion to address this. Lines 530 – 532:

“The recommended management of genital and anorectal gonorrhoea infections in Australia is a dual therapy of Ceftriaxone 500mg and Azithromycin 1g (Azithromycin 2g for pharyngeal infection) (60)”

4. Lines 285 and 293: Why couldn’t NG-MAST or NG-STAR be assigned in these cases?

2,483 isolates could not be assigned an NG-MAST profile. This was because NG-MASTER identified one or both genes (*tpbB* and *porB*) as having (a) novel full length alleles similar to a known allele, (b) a partial match to a known allele, (c) a missing allele or (d) multiple alleles. The allele results for the two genes can be found in the Supplementary Dataset.

For the 3 isolates that could not be assigned an NG-STAR profile, they were not able to be typed at *porB* gene which is required for an overall profile.

5. Both “genomic clusters” and “transmission clusters” seem to be used interchangeably. I suggest sticking with one term. Using transmission clusters may be making some assumptions about some of the clusters where perhaps there are not sufficient epi links to infer transmission.

We have changed “*transmission clusters*” to “*genomic clusters*” throughout.

6. Lines 434-435: Any thoughts on why persistence would be associated with azithromycin susceptibility (especially if azithromycin is being used as a treatment)?

We have elaborated on this point in the discussion in lines 536 – 544:

“Susceptibility to azithromycin, a current recommended therapeutic, was associated with persistence, further highlighting the complex interplay between antimicrobial susceptibility and transmission. This finding aligns with existing research demonstrating a fitness cost associated with antibiotic resistance. For instance, cefixime-resistant strains of N. gonorrhoeae have been shown to cause approximately half the number of secondary infections compared to susceptible strains (59). Similarly, azithromycin-resistant Staphylococcus aureus strains exhibit reduced fitness compared to azithromycin-sensitive strains (60). These findings collectively suggest that resistance may incur a fitness disadvantage, which could reduce the transmission potential of resistant strains and that alternate factors such as host-pathogen interactions or behavioural dynamics may play more central roles in persistent chains of transmission.”

7. Lines 558-559: In the Figure 4 legend it says “unique isolates for each gene”, however in the figure and results it says “unique alleles per gene”, so it may be a typing error where "isolates" needs to be changed to "alleles".

This has been corrected.

8. Supplementary Figure 7: When multiple isolates from a patient or multiple colonies were sequenced for an isolate, how were these chosen? Are they from different anatomic sites? Did this practice occur on a convenience basis or were they systematically chosen?

We have included a sentence at 309 – 211 to clarify this point:

“In total, 5,881 genomes (one samples collected per individual, per day; this was randomly selected for within-individual samples) were included for clustering and epidemiological analyses”

We elected to randomly select one genome to include when there was multiple collected from the same patient at the same time to avoid biasing by favouring any particular site of collection.

9. Supplementary Figure 10: Was this type of SNP alignment done for all of the clusters?

This kind of SNP alignment was done for all clusters with 5 or more genomes collected between 2019 and 2021 as part of the effective reproductive number calculation. We elected to include a schematic of this cluster to demonstrate the benefit of using a cgMLST scheme for clustering, as opposed to a SNP based method that would be heavily influenced and biased by recombination events. Please see the response to Reviewer 2 Comment 26.

Comments and suggestions specific to the phylogenomic and phylodynamic methods:

10. 18% of the reported cases in Victoria were sequenced, which is a very good representative sample to perform population transmission dynamic analysis and infer trends especially before and after COVID-19 pandemic. The dataset is unique and most of analysis are performed appropriately. One major concern I have is that authors have not systematically removed the recombination events while performing their phylogenomic and Phylodynamic analysis. I have made a few suggestions to improve the analysis results and made some suggestions to perform some additional comparative analysis to assess whether different methods provide similar trends.

Thank you for the comments, we aim to address these below.

11. Lines 165 – 174: Authors concatenated all the cgMLST gene (n=1648 genes) alignments and generated a whole genome core-SNP alignment, which was used for the ML phylogenetic analysis. This inferred ML tree was used for molecular dating. A root to tip correlation is not shown but it is required for timed phylogenetic analysis. I see a regression plot in Figure 1C, which appears to have a positive correlation, but it would be great to provide the estimated correlation.

We used a fixed rate estimate (4.54×10^{-6}) based on the literature (see response to Reviewer 1 Comment 2) and have shown a positive correlation between pairwise allelic difference and time between sample collection in Figure 1C. The slope of this line was 2.597362 allele differences per year as stated in lines 384 – 385 of the manuscript.

Nonetheless, a root-to-tip regression of the concatenated cgMLST gene maximum likelihood phylogenetic tree was generated using Clockor2 (<https://clockor2.github.io/>), where the R^2 of the slope was positive (0.150) and the residual mean squared was 5.649×10^{-7} .

12. Considering the amount of recombination events in GC, it would be beneficial to the readers to see whether the branch lengths as well as molecular dating of ancestral events will be similar to cgMLST phylogenetic analysis if the whole genome alignment with 5881 genomes was generated using snippy followed by recombination detection using gubbins and use the resultant SNP alignment (outside of the recombinant regions) for ML phylogenetic analysis with ascertainment bias correction. The ascertainment bias correction imposed due to the usage of a SNP alignment with no recombination events will alleviate authors concerns that the use of a strict core genome alignment potentially results in isolates being classified as more closely related than they would be (as mentioned by authors in the discussion section (lines 396 – 399)). I totally understand that it is inappropriate to do recombination detection and masking using gubbins on a concatenated core-gene alignment. A ML phylogenetic tree figure with all the major MLSTs mapped to it along with some important AMR markers will be a good addition to the manuscript.

We thank the reviewer for this comment. To address this, the trimmed paired end reads were aligned to the NCCP11945 reference genome using Snippy (v4.3.5), requiring a minimum of ten supporting reads and a variant frequency of 0.9 or greater. Recombination filtering was performed using Gubbins (v2.4.1) with default settings and the full Snippy pseudoalignments as input. Following Gubbins, a core SNP alignment was generated using snp-sites (v1) and the Gubbins filtered alignment as input with the -c flag. The number of constant sites from the whole genome pseudoalignment was also calculated using snp-sites with the -C flag (v1). A ML phylogenetic tree was inferred using IQ-tree (v2.0.3), with the best-fitting nucleotide substitution model chosen based on the lowest BIC and the number of constant sites specified. Molecular dating of ancestral events was performed using the least-squares dating (LSD) software (v0.3), with the whole dataset maximum likelihood phylogeny generated here used as an input.

Using this method, the core SNP alignment consisted of 8,842 core sites (polymorphic/variants that are present in all samples). While using a core SNP alignment can be used to build a high-resolution phylogeny in many cases and has been the gold standard approach in bacterial phylogenetics for the past decade, when applied to a large and diverse dataset as in this case, it

can result in a shrinking of informative sites, and a less resolved phylogeny. *N. gonorrhoeae* is a very diverse species with much recombination, therefore the number of sites conserved across various samples is smaller. For example, across the full whole genome pseudoalignment, there is a minimum of 6% N sites (134,866 bp) for any genome. As these N sites will be dispersed mostly randomly across the genome, the chances of any site having at least one N in at least one sample is high and means that site will be excluded from the core SNP alignment, even if it is informative to the phylogeny. One way to increase the number of core SNP sites is to remove genomes with a high proportion of N sites from the alignment and analysis. In this case, 1,184 genomes have more than 10% N sites. Excluding these would remove 20% of isolates from the phylogeny and clustering analysis – decreasing our sampling proportion and introducing a high level of uncertainty into our clustering. As a result, we opted to use a cgMLST method, a much more permissive way of comparing relatedness across very diverse genomes in a large dataset.

The concern that using a strict core results in isolates potentially being classified as more closely related than they would be stems from the principle of using a static SNP threshold to define transmission. For example, a common SNP threshold of 10 SNPs would result in much more permissive clustering using a core SNP alignment of one hundred sites compared to using the same threshold on a larger core of thousands, as the threshold represents a fraction of the total sites.

Regardless, we have generated a recombination filtered core genome SNP ML phylogeny. We can see that the overall population structure is mostly conserved, however the resolution at recent evolutionary events is much reduced when compared to the concatenated cgMLST phylogeny.

SNP tree:

cgMLST concatenated tree:

Additionally, we generated a timed phylogeny using LSD and the ML SNP tree from above. This resulted in a phylogeny with a tMRCA of 1845.73 compared to the cgMLST tree root of 1873.13.

We agree it would be beneficial to the readers to include a ML phylogeny with MLST and AMR metadata and have now included this as Supplementary Figure 8. We also agree that this

is a useful comparison and have included this entire analysis in the Supplementary Appendix and in the Supplementary Analyses detailed in the GitHub associated with this publication (https://github.com/mtaouk/Neisseria_gonorrhoeae_transmission_Australia/tree/main/Supplementary_analyses/SNP_alignment).

13. It would be great if the authors could also provide the SNP differences from the above suggested recombination removed SNP alignment inferred from gubbins and provide a comparison to the reported allele differences between individuals, within individuals, within sites and in paired couples (Lines 309 – 311).

Following on from the analysis detailed in our previous response, we calculated the pairwise SNP distances across the dataset from the recombination filtered core SNP alignment using `snp-dists` (v1). The median pairwise SNP distance between individuals was 122 (range 0 to 769), within individuals was 0 (range 0 to 211), within sites was 0 (range 0 to 0) and in paired couples was 0 (range 0 to 0). We find that using a strict core SNP alignment of 8,842 sites, decreases the resolution of relatedness between any isolates and decreases the utility of our calibration isolates in determining a threshold, as the maximum pairwise SNP distance is 0.

We agree this is a useful comparison and have also included this analysis in the Supplementary Appendix and in the Supplementary Analyses detailed in the GitHub associated with this publication: (https://github.com/mtaouk/Neisseria_gonorrhoeae_transmission_Australia/tree/main/Supplementary_analyses/SNP_alignment).

14. Also, a comparison of the estimated allelic distance per week to that of the SNPs distance per week (lines 313 to 315) will be very valuable to the entire GC genomics community.

We have estimated an evolutionary rate of 4.995×10^{-2} allelic differences per week or 2.6 allelic differences per year. This is in line with the previous studies estimating the evolutionary rate of *N. gonorrhoeae* in SNP distances. Please see the response to Reviewer 1 Comment 2 citing these previous publications.

15. Lines 196 – 204: Glad to see that the authors generated separate ML trees for their clusters using snippy generated whole genome alignment, which is highly appreciated but the way of arbitrarily removing recombination regions based on the longest branch lengths is not acceptable. I would strongly recommend implementing recombination correction using gubbins on those trees displaced (supplementary figure 15) so that mostly all the clusters will be included in the analysis. Gubbins will be very efficient in recombination detection within an alignment of genetically similar isolates. Please update the timed trees as well (supplementary figure 16).

Thank you for this suggestion. We have now incorporated Gubbins into our analysis. Please see our response to Reviewer 1 Comment 2.

16. It would be great if the top MLST sequence type is indicated for each of the cluster trees in supplementary figures 15 and 16). There is no way to correlate cluster designations mentioned in figure 3 to the cluster names in supplementary figures 15 and 16.

Thank you for this suggestion, this has now been included in what is now Supplementary Figure 15.

17. Lines 206 – 233. It would be great if the authors could implement TransPhylo (Didelot et al. 2017) and estimate effective reproductive numbers and check whether the estimates are similar. Transphylo has been implemented previously in GC (Joseph et al. 2022; Osnes et al. 2020). Transphylo will also estimate the potential unsampled isolates in the transmission chain (see Figure S5 in (Joseph et al. 2022)) and it will be ideal to run it on the timed phylogenies estimated for the clusters.

Thank you for this suggestion. Transphylo does include models that would estimate similar parameters, but estimating individual parameters for each transmission cluster, as in Joseph et al. (2022) is unfeasible here. In particular, as shown in Supplementary Fig 15, some of our transmission clusters only have 5 sequences and very short branches. Our hierarchical approach allows such small clusters to contribute to weighted average of parameters and to leverage their signal via hyperprior distributions.

18. Authors could also estimate the population trajectories estimated by modeling the effective population size over time for each of their clusters using the currently implemented birth-death skyline process similar to shown in (Joseph et al. 2022 Figure S2). This will clearly indicate the population trajectories of these clusters before, during COVID 19 restrictions and after the removal of COVID 19 restrictions.

As explained in our previous point, this approach is inappropriate because we are working with some very small clusters. Applying a demographic model individually to these clusters could result in considerable overfitting.

REVIEWERS' COMMENTS

Reviewer #1 (Remarks to the Author):

The revision by Taouk et al. has addressed all of my concerns with the previous version of the manuscript. The authors have re-analyzed their data using methods that identify and mask recombination events, investigated the impact of the thresholds used on the associations between susceptibility and persistence, and included additional supplementary data/code in supplementary tables or on GitHub.

I do think it would be useful to include the additional analysis on breakpoints/MICs and persistence that are in the rebuttal document in the supplement. Given the differences in susceptibility breakpoints across organizations/geographic regions, the result that using continuous MIC values in the analysis identifies the same variables as significant may be of interest to readers outside of Australia.

Reviewer #1 (Remarks on code availability):

The GitHub repository contains a detailed README with all commands used for the analyses presented in this manuscript. Much of the code is actually written into this README rather than in individual scripts, a Rmarkdown/Jupyter notebook, or a Nextflow/Snakemake pipeline, which would probably make it more difficult to reproduce the results of the paper (e.g. would require copying and pasting rather than just running a script).

Reviewer #2 (Remarks to the Author):

Congratulations on this comprehensive work, and thank you for addressing all my comments. It has been a pleasure to review your work.

I have only two minor comments:

1. This would make the genotyping more complete. I bring this up because two reviewers commented on it, and readers will most likely question it as well. If the authors choose a second assembler (Spades, Skesa, Velvet, or other) and reassemble the genomes from the raw data, that would most likely solve many of the genomes that couldn't be assigned one or several genotyping genes.

2. Regarding my previous comment on NG-MAST (#8), to be clear: was ngmaster used on the PubMLST (NG-MAST v2.0) database? If so, I want the authors to be aware of the differences in trimming porB and tpbB. Ngmaster was designed for NG-MAST v1.0 (no longer maintained). Over the past couple of years, the curation of NG-MAST has changed. PorB starts at TTGAA as in NG-MAST v1.0, but instead of being 490 bps as NG-MAST v1.0 and as ngmaster assumes, it has to end with CGCACAAC to be compatible with NG-MAST v2.0. TpbB starts at CGTCTG (as in NG-MAST v1.0) but with a more complicated end site AAAACTGC or AAAA???? - this is in contrast to NG-MAST v1.0 where it was 390 bps. The authors could submit all their sequences to the PubMLST database to compare and confirm their NG-MAST STs, and I think any discrepancies can be solved with curators' support. The author of ngmaster could very well have updated ngmaster to handle this, but I know for a fact that the trimming has been quite complicated and required some manual handling recently.

Reviewer #2 (Remarks on code availability):

1. Include how the post-gubbins with recombinations masked was obtained? Specifically, how was "core.full.Gubbins.aln" obtained?

Reviewer #3 (Remarks to the Author):

I appreciate the opportunity to review the revised manuscript by Taouk et al. The authors appear to have carefully addressed the comments and suggested edits provided from the first submission, and I appreciate the additional analyses that were added. Comparison of new versus existing methods is important to advancing the field. There are a few minor comments remaining regarding the following:

-Response to comment #12: Thanks for performing a core SNP-based phylogenetic analysis and comparing it with the tree inferred using cgMLST. I am glad to see that the overall population structure was preserved across both methods. I understand that *Ng* is a very diverse species and if you take all of your 5000+ genomes and do recombination masking with Gubbins your core genome or core SNP sites will be reduced as it will remove both the ancestral and recent recombination events. I still believe using a core SNP phylogenetic inference with ascertainment bias correction is still valid and a systematic approach to perform a global phylogenetic analysis to understand the overall population structure compared to the cgMLST.

Regarding authors' concern that using a strict core result in isolates potentially being classified as more closely related than they would be stems from the principle of using a static SNP threshold to define transmission and also not efficient in inferring recent evolutionary events – It is not recommended to extract SNP differences from a whole genome alignment with 5000+ genomes to infer transmission events. For inferring transmission events, clade or cluster specific alignments should be generated and then recombination regions should be removed/masked and only the vertically inherited (as opposed to horizontally inherited SNPs) SNPs should be extracted as a metric for genetic differences between 2 or more isolates. When you perform a clade/cluster/core geno group alignment by mapping your reads to a genetically closer reference genome you will be getting a lot of genetically informative insights compared to what you get based on cgMLST and the former will definitely provide greater resolution. Please make this point clear in the discussion section as well as in the supplementary analysis/materials.

-Response to comments #17 and #18: Regarding performing TransPhylo- I understand authors' point. They could also do this on a few important large clusters (at least 1 or 2), and I think the dataset they have is ideal for a TransPhylo analysis as it models and accounts for the unsampled individuals as well. I will defer this to the editor.

-Supplemental Figure 15: There does not appear to be mean SNP differences (after removal of recombination events) for each of the 90+ clusters shown in Supplemental Figure 15. Adding this information to the figure could help with interpretation and comparison.

REVIEWERS' COMMENTS

Reviewer #1 (Remarks to the Author):

The revision by Taouk et al. has addressed all of my concerns with the previous version of the manuscript. The authors have re-analyzed their data using methods that identify and mask recombination events, investigated the impact of the thresholds used on the associations between susceptibility and persistence, and included additional supplementary data/code in supplementary tables or on GitHub.

We thank the reviewer for the positive feedback.

I do think it would be useful to include the additional analysis on breakpoints/MICs and persistence that are in the rebuttal document in the supplement. Given the differences in susceptibility breakpoints across organizations/geographic regions, the result that using continuous MIC values in the analysis identifies the same variables as significant may be of interest to readers outside of Australia.

We thank the reviewer for this suggestion and have included this analysis in the Supplementary material and accompanying GitHub for replication.

Reviewer #1 (Remarks on code availability):

The GitHub repository contains a detailed README with all commands used for the analyses presented in this manuscript. Much of the code is actually written into this README rather than in individual scripts, a Rmarkdown/Jupyter notebook, or a Nextflow/Snakemake pipeline, which would probably make it more difficult to reproduce the results of the paper (e.g. would require copying and pasting rather than just running a script).

We appreciate the reviewer's suggestion, although believe the present arrangement achieves an appropriate balance between ease of use and reproducibility. The inclusion of all relevant code in the publicly available README allows users to easily understand and modify the steps according to their specific datasets and research questions. Alternative formats as suggested can easily be prepared from the information made available.

Reviewer #2 (Remarks to the Author):

Congratulations on this comprehensive work, and thank you for addressing all my comments. It has been a pleasure to review your work.

We thank the reviewer for the positive feedback.

I have only two minor comments:

1. This would make the genotyping more complete. I bring this up because two reviewers commented on it, and readers will most likely question it as well. If the authors choose a second

assembler (Spades, Skesa, Velvet, or other) and reassemble the genomes from the raw data, that would most likely solve many of the genomes that couldn't be assigned one or several genotyping genes.

We thank the reviewer for this suggestion. As mentioned in the manuscript, assignment is highly complete for NG-Star (5878/5881; >99.95%) and MLST (5837/5881; 99.25%), given we are using a modern and appropriate genome assembler (*shovill*). This approach improves on the *spades* assembler suggested by incorporating read preprocessing and other steps. As such, further reassembly will likely not improve assignment and may make interpretation and reproduction of our methods more complicated. Please see our response to Reviewer 1 comment 2 regarding NG-MAST typing.

2. Regarding my previous comment on NG-MAST (#8), to be clear: was ngmaster used on the PubMLST (NG-MAST v2.0) database? If so, I want the authors to be aware of the differences in trimming *porB* and *tpbB*. Ngmaster was designed for NG-MAST v1.0 (no longer maintained). Over the past couple of years, the curation of NG-MAST has changed. *PorB* starts at TTGAA as in NG-MAST v1.0, but instead of being 490 bps as NG-MAST v1.0 and as ngmaster assumes, it has to end with CGCACAACT to be compatible with NG-MAST v2.0. *TpbB* starts at CGTCTG (as in NG-MAST v1.0) but with a more complicated end site AAAACTGC or AAAA???? - this is in contrast to NG-MAST v1.0 where it was 390 bps. The authors could submit all their sequences to the PubMLST database to compare and confirm their NG-MAST STs, and I think any discrepancies can be solved with curators' support. The author of ngmaster could very well have updated ngmaster to handle this, but I know for a fact that the trimming has been quite complicated and required some manual handling recently.

As this typing stage was not relevant to the study findings, and as the standard database and typing approach appear incompatible, we have removed reference to it for clarity. We note that the Australian read data and assemblies have been made publicly available, and so this can be pursued by others interested in multi-antigen typing including automatic NGMAST typing on the public Pathogenwatch page.

Reviewer #2 (Remarks on code availability):

1. Include how the post-gubbins with recombinations masked was obtained? Specifically, how was "core.full.Gubbins.aln" obtained?

Thank you for this suggestion. We used the `filtered_ploymorphic_sites.fasta` file output by Gubbins. We have renamed the files in the README to be clearer:

4. Recombination filtering

Recombination filtering was performed using Gubbins (v2.4.1) with default settings for all clusters with the full whole genome psuedoalignments as input:

```
run_gubbins.py --threads 10 full_alignments/group_14_snippy.fasta
```

For each cluster gubbins filtered alignment, a SNP alignment was generated:

```
snp-sites -c -o group_14_snippy.filtered_ploymorphic_sites.fasta group_14_snippy_gubbins.fasta
```

Reviewer #3 (Remarks to the Author):

I appreciate the opportunity to review the revised manuscript by Taouk et al. The authors appear to have carefully addressed the comments and suggested edits provided from the first submission, and I appreciate the additional analyses that were added. Comparison of new versus existing methods is important to advancing the field. There are a few minor comments remaining regarding the following:

We thank the reviewer for the positive feedback.

-Response to comment #12: Thanks for performing a core SNP-based phylogenetic analysis and comparing it with the tree inferred using cgMLST. I am glad to see that the overall population structure was preserved across both methods. I understand that Ng is a very diverse species and if you take all of your 5000+ genomes and do recombination masking with Gubbins your core genome or core SNP sites will be reduced as it will remove both the ancestral and recent recombination events. I still believe using a core SNP phylogenetic inference with ascertainment bias correction is still valid and a systematic approach to perform a global phylogenetic analysis to understand the overall population structure compared to the cgMLST.

Regarding authors' concern that using a strict core result in isolates potentially being classified as more closely related than they would be stems from the principle of using a static SNP threshold to define transmission and also not efficient in inferring recent evolutionary events – It is not recommended to extract SNP differences from a whole genome alignment with 5000+ genomes to infer transmission events. For inferring transmission events, clade or cluster specific alignments should be generated and then recombination regions should be removed/masked and only the vertically inherited (as opposed to horizontally inherited SNPs) SNPs should be extracted as a metric for genetic differences between 2 or more isolates. When you perform a clade/cluster/core geno group alignment by mapping your reads to a genetically closer reference genome you will be getting a lot of genetically informative insights compared to what you get based on cgMLST and the former will definitely provide greater resolution. Please make this point clear in the discussion section as well as in the supplementary analysis/materials.

We thank the reviewer for their feedback. We have included a section in the Supplementary Analysis to address these points:

“We acknowledge that inferring transmission events could benefit from clade- or cluster-specific alignments. By generating alignments specific to a clade or core genome group and mapping reads to a genetically closer reference genome, it would be possible to obtain more genetically informative insights compared to what is obtained based on cgMLST.”

-Response to comments #17 and #18: Regarding performing TransPhylo- I understand authors' point. They could also do this on a few important large clusters (at least 1 or 2), and I think the dataset they have is ideal for a TransPhylo analysis as it models and accounts for the unsampled individuals as well. I will defer this to the editor.

Applying a phylodynamic approach to a few larger clusters, although potentially interesting for other reasons, would not directly address the research aims of this specific project. As such, we have opted not to include this suggested analysis in this particular study. As mentioned previously however, sequence data have been made publicly available, with colleagues in the field being free to pursue these research ideas.

-Supplemental Figure 15: There does not appear to be mean SNP differences (after removal of recombination events) for each of the 90+ clusters shown in Supplemental Figure 15. Adding this information to the figure could help with interpretation and comparison.

We thank the reviewer for this suggestion and have now included this information in Supplementary Table 3, retaining clarity on Supplementary Figure 15.